# First Observation of Mercury Species on an Important Water Vapor Channel in the Southeast Tibetan Plateau

Huiming Lin[1], Yindong Tong[2*], Chenghao Yu[1], Long Chen[3], Xiufeng Yin[4], Qianggong Zhang[5,6], Shichang Kang[4,6], Lun Luo[7], James Schauer[8,9], Benjamin de Foy[10], Xuejun Wang[1**]

**Affiliations**

1. MOE Laboratory of Earth Surface Processes, College of Urban and Environmental Sciences, Peking University, Beijing 100871, China;

2. School of Environmental Science and Engineering, Tianjin University, Tianjin, 300072, China;

3. School of Geographic Sciences, East China Normal University, Shanghai 200241, China;

4. State Key Laboratory of Cryospheric Science, Northwest Institute of Eco-Environment and Resources, Chinese Academy of Sciences, Lanzhou 730000, China

5. Key Laboratory of Tibetan Environment Changes and Land Surface Processes, Institute of Tibetan Plateau Research, Chinese Academy of Sciences, Beijing, 100101, China;

6. CAS Center for Excellence in Tibetan Plateau Earth Sciences, Beijing, 100085, China;

7. South-East Tibetan plateau Station for integrated observation and research of alpine environment, CAS

8. Department of Civil and Environmental Engineering, University of Wisconsin-Madison, Madison, WI, USA;

9. Wisconsin State Laboratory of Hygiene, University of Wisconsin-Madison, WI, USA;

10. Department of Earth and Atmospheric Sciences, Saint Louis University, St. Louis, MO, 63108, USA

*Correspondence to*:

*Yindong Tong, Tianjin University, Tianjin, China, Email at: yindongtong@tju.edu.cn;

**Xuejun Wang, Peking University, Beijing, China, Email at: wangxuejun@pku.edu.cn

**Abstract**

The Tibetan Plateau is generally considered to be a significantly clean area owing to its high altitude; however, the transport of atmospheric pollutants from the Indian subcontinent to the Tibetan Plateau has influenced the Tibetan environments. Nyingchi is located at the end of an important water vapor channel.

In this study, continuous monitoring of gaseous elemental mercury (GEM), gaseous oxidized mercury (GOM), and particle-bound mercury (PBM) was conducted in Nyingchi from March 30 to September 3, 2019, to study the influence of the Indian summer monsoon (ISM) on the origin, transport and behavior of Hg. The GEM and PBM during the preceding Indian summer monsoon (PISM) period ($1.20\pm0.35$ ng m$^{-3}$, and $11.4\pm4.8$ pg m$^{-3}$ for GEM and PBM, respectively) were significantly higher than those during the ISM period ($0.95\pm0.21$ ng m$^{-3}$, and $8.8\pm6.0$ pg m$^{-3}$), the GOM during the PISM period ($13.5\pm7.3$ pg m$^{-3}$) was almost at the same level with that during the ISM period ($12.7\pm14.3$ pg m$^{-3}$). The average GEM concentration in the Nyingchi region was obtained using passive sampler as $1.12\pm0.28$ ng m$^{-3}$ (from April 4, 2019 to March 31, 2020). The GEM concentration showed that the sampling area was very clean compared to other high-altitude sites. The GEM has several patterns of diurnal variation during different periods. Stable high GEM concentrations occur at night and low concentrations occur at afternoon during PISM, which may be related to the nocturnal boundary layer structure. High values occurring in the late afternoon during the ISM may be related to long-range transport. Low concentrations of GEM observed during the morning in the ISM may originate from vegetation effects. The results of the trajectory model demonstrate that the sources of pollutants at Nyingchi are different with different circulation patterns. During westerly circulation in PISM period, pollutants mainly originate from central India, northeastern India, and central Tibet. During the ISM period, the pollutants mainly originate from the southern part of the SET site. The strong precipitation and vegetation effects on Hg species during the ISM resulted in low Hg concentrations transmitted to Nyingchi during this period. Further, principal component analysis showed that long-distance transport, local emissions, meteorological factors, and snowmelt factors are the main factors affecting the local Hg concentration in Nyingchi. Long-distance transport factor dominates during PISM and ISM3, while local emissions is the major contributor between PISM and ISM3. Our results reveal the Hg species distribution and possible sources of the most important water vapor channel in the Tibetan Plateau, and could serve a basis for further transboundary transport flux calculations.

## 1. Introduction

Mercury (Hg) is classified as a hazardous pollutant because it is bio-accumulative and toxic (Mason et al., 1994; Mason et al., 1995). Generally, atmospheric Hg can be categorized into three major types:

gaseous elemental mercury (GEM), gaseous oxidized mercury (GOM), and particle-bound mercury
(PBM) (Selin, 2009). The stable chemical properties of GEM coupled with its long atmospheric lifetime
(approximately 0.3 to 1 year) makes GEM an important global pollutant (Selin, 2009; Travnikov et al.,
2017). In contrast, GOM and PBM are easily removed from the atmosphere through chemical reaction
and deposition because of their chemical activity and water solubility, and could therefore bring
significant impacts to the local environment (Lindberg and Stratton, 1998; Seigneur et al., 2006). Both
GOM and PBM have complex fundamental physicochemical properties and may have complicated
relationships with other regional pollutants (Gustin et al., 2015). Understanding, identifying, and
characterizing Hg sources and their global and regional transport mechanisms is crucial for global
atmospheric Hg control and health effects research (UNEP, 2018). Since 2013, the Minamata Convention
was established to control the global mercury pollution (UNEP, 2013a). Monitoring atmospheric Hg is
an important prerequisite for implementing the convention. Currently, several Hg monitoring networks
and studies have been established to better understand atmospheric Hg cycling. The Atmospheric
Mercury Network (AMNet; Gay et al., 2013), the Global Mercury Observation System (GMOS;
Sprovieri et al., 2013;Sprovieri et al., 2016), the Canadian Atmospheric Mercury Network (CAMNet;
Kellerhals et al., 2003) and the Arctic Monitoring Assessment Programme(AMAP;
https://mercury.amap.no/) are the main monitoring networks operating in North America and Europe,
and the majority of them only monitor GEM concentrations (Gay et al., 2013; Sprovieri et al., 2013;
Sprovieri et al., 2016; Kellerhals et al., 2003). Researchers worldwide have also contributed to
monitoring the data from different regions (Gustin et al., 2015; Jiang and Wang, 2019; Stylo et al., 2016).
In China, which has received more attention, there are no reported atmospheric Hg observation networks,
but there has been considerable monitoring work by different organizations (Fu et al., 2012b; Fu et al.,
2008; Fu et al., 2016a; Fu et al., 2019; Fu et al., 2016b; Liu et al., 2011; Feng and Fu, 2016; Feng et al.,
2013; Wang et al., 2015b; Lin et al., 2019; Hu et al., 2014; Ci et al., 2011; Duan et al., 2017; Liu et al.,
2002; Yin et al., 2018; Yin et al., 2020). However, there exists some gaps in understanding the sources
and transport of atmospheric Hg in some remote areas, especially in harsh environmental areas where
performing monitoring is difficult.
Considering that GEM can be transported globally over long distances and that the transport
distances of GOM and PBM vary greatly in different environments, atmospheric Hg concentration
monitoring may not directly reflect the intensity of regional atmospheric Hg emissions. Our previous
study of the Qomolangma National Nature Preserve (QNNP) (Lin et al., 2019) demonstrated that the Hg
emitted from India can cross the Himalayas to reach the Tibetan Plateau. Further research on the
transboundary transport of Hg should be conducted to better understand the transport mechanisms. This
is particularly true in Asia, where the environmental pollution is generally severe. China and India are
reported to be the world's largest consumers of coal (BP Statistical Review of World Energy, 2018).
Considering that coal is the largest emission source of Hg in the atmosphere (approximately 86% of fuel-
related atmospheric Hg emissions come from fuel combustion (Chen et al., 2016)), both China and India
have great Hg emission potential. South Asia, and East and Southeast Asia accounted for 10.1% and 38.6%
of global emissions of mercury, respectively (UNEP, 2018; Zhang et al., 2015b). Further research on
pollutant transport in Asia should be conducted to support policy development and responsibility
allocation.
The Tibetan Plateau, with an average elevation of more than 4,000 m above sea level, is a natural
barrier between inland China and the Indian subcontinent (Qiu, 2008; Lin et al., 2019). In the southern
part of the Tibetan Plateau, the Himalayas, with an average altitude of 6,000 m, can serve as a solid
barrier to pollutant transport. However, this barrier cannot completely block the transboundary
transportation of pollutants according to previous studies. The transboundary and long-distance transport
of pollutants across the Himalayas has attracted considerable attention (Wang et al., 2018; Zhang et al.,
2015a; Yang et al., 2018; Li et al., 2016; Feng et al., 2019; Zhu et al., 2019). Several studies have shown
that the transboundary intrusion of atmospheric pollutants through the Himalayas on the Tibetan Plateau
is crucial for many pollutants (Yang et al., 2018; Li et al., 2016; Zhang et al., 2015c; Pokhrel et al., 2016;
Lin et al., 2019). Zhang et al. (2017) studied short-lived reactive aromatic hydrocarbons and indicated
that the cut-off low system that have lower altitude in the Himalayas is a major pathway for long-distance
transport of aromatic hydrocarbons in the Tibetan Plateau. Persistent organic pollutants have been
reported to be transported to the interior of the Tibetan Plateau by traveling along valleys or across ridges
(Gong et al., 2019a). The transport of aerosols and organic pollutants along the most important water
vapor channel, the Yarlung Zangbu/Brahmaputra Grand Canyon (hereafter referred to as the YZB Grand
Canyon), has been observed (Wang et al., 2015a; Sheng et al., 2013).
In the case of atmospheric Hg, monitoring in marginal areas depicted the basic spectrum of
atmospheric Hg in the Tibetan Plateau. Monitoring of atmospheric Hg at Shangri-La, Nam Co,
Qomolangma, Mt. Gongga, Mt. Waliguan and Mt. Yulong have illustrated atmospheric Hg
concentrations and transport patterns in the Tibetan Plateau from multiple perspectives, all of which also
indicate the effects of transboundary transport on the atmospheric Hg concentrations in the Tibetan
Plateau (Zhang et al., 2015a; Yin et al., 2018; Lin et al., 2019; Fu et al., 2008; Fu et al., 2012a; Wang et
al., 2014). For example, our previous study in the QNNP, on the southern border of the Tibetan Plateau,
proved that atmospheric Hg from the Indian subcontinent can be transported across high-altitude
mountains, and directly to the Tibetan Plateau under the action of the Indian monsoon and local glacier
winds (Lin et al., 2019). Studies of water vapor mercury and wet deposition of Hg in cities such as Lhasa
have demonstrated higher concentrations of Hg species (Huang et al., 2015; Huang et al., 2016b; Huang
et al., 2016a). But the monitoring of atmospheric Hg speciation is still rare. However, to the best of our
knowledge, the monitoring of the passage of atmospheric Hg in the main water vapor channel—the YZB
Grand Canyon, into the Tibetan Plateau has not been conducted. Through the water vapor and airflow
channel, air masses carrying large amounts of water vapor as well as pollutants may enter Tibet, resulting
in heavy precipitation during the monsoon season. Huang et al. (2015) reported that the total Hg wet
deposition in Nyingchi, located in the YZB Grand Canyon, was lower than that in other Tibetan Plateau
regions, and the concentration was lower in the monsoon season than in the non-monsoon season. As an
important transport channel for summer monsoon moisture into China (Xu et al., 2020; Feng and Zhou,
2012; Yang et al., 2013), the amount of water vapor transported into Tibet through this channel is
considerable, and the transport of pollutants needs further investigation.

137        In this study, we set up high time resolution Hg species monitoring in Nyingchi, southeastern

Tibetan Plateau, covering both PISM and ISM periods. Hg passive sampling was also applied to cover
the monitoring of the entire year. To the best of our knowledge, this is the first monitoring study of
atmospheric Hg species in the most important water vapor channel of the Tibetan Plateau. To better
identify the sources of Hg pollution and potential pollution areas, we combined real-time GEM
monitoring data with backward trajectory analysis, and a follow-up cluster analysis of the trajectories.
We also collected other pollutant concentrations and rainfall data near the monitoring station during the
same period to better analyze the sources and transport characteristics of Hg. By combining the real-time
monitoring data and model simulations, we attempted to better characterize the process of Hg entering
Tibet through the water vapor channel, which could allow researchers to further analyze the transport of
Hg from the Indian subcontinent into Tibet and provide scientific support for managerial decision making.
**2. Materials and methods**
**2.1 Atmospheric Hg monitoring site**

150       Atmospheric Hg monitoring was performed at the South-East Tibetan Plateau Station for Integrated

Observation and Research of Alpine Environment (SET station, Figure 1) in Nyingchi, Tibet, China. The
SET station is located in the southeastern part of the Tibetan Plateau (29°45′59N, 94°44′16E, 3263 m
a.s.l.), in a water vapor transportation channel, from the Ganges River Plain to the Tibetan Plateau. The
meteorological factors at Nyingchi are mainly controlled by westerly winds (from September to April)
and ISM (from May to August), exhibiting sharp seasonal variations (controlling date was decided
according to Indian Monsoon Index, Figure S1). The average annual air temperature is 5.6 °C, the
average air temperature during PISM and ISM periods are 6.0 °C and 12.0 °C, respectively. The
Tibetan Plateau is generally a moisture sink in summer (Feng and Zhou, 2012; Xu et al., 2020), with
climatological moisture originating from the Indian Ocean and the Bay of Bengal intruding into the center
of the Tibetan Plateau along the water vapor channels. The average annual precipitation is approximately
700-1000 mm at the SET station, much higher than the annual precipitation in Tibet (596.3 mm in 2019).
The precipitation at the SET station is 47.7 mm during the period of PISM, and is 528.5 mm during the
period of ISM in 2019. During the westerly period, the air masses are mainly from mid-latitude inland
areas with less water vapor, while during the ISM period, a large amount of water vapor from the Indian
Ocean enters Tibet. The precipitation begins at the foot of the YZB Grand Canyon and is sustained along
with the canyon into Tibet (Gong et al., 2019b), and the precipitation in the downstream Motuo County
is more than twice that of the Nyingchi area (Ping and Bo, 2018). The unique geomorphological
conditions and the effect of the strong monsoon have resulted in a unique high-altitude distribution
pattern of various biomes and vegetation in the area. Interactions between terrestrial ecosystems and
atmosphere have contributed to the development of diverse biomes and distinctive vegetation elevation
distribution patterns from tropical rainforests to boreal forests and tundra. The SET station is 75 km from
Bayi Town, where the capital of Nyingchi Prefecture is located, and 480 km from Lhasa, which is the
capital city of the Tibet Autonomous Region. Owing to the high altitude and harsh living environment,
the permanent population in Tibet is extremely small and only a few local pollutant emission sources
have been observed (UNEP, 2013b; UNEP, 2018).
**2.2 GEM, GOM and PBM active monitoring**
Real-time continuous measurements of GEM, GOM, and PBM concentrations were carried out
using Tekran Model 2537B, 1130, and 1135 instruments (Tekran Inc., Toronto, Canada) at the SET station
from March 30 to September 3, 2019, which could show the diurnal and daily changes in atmospheric
Hg concentration in detail. During the operation of the Tekran instruments, the sampling inlet was set at
~1.5 m above the instrument platform (shown in Figure S2). Considering the high altitude at which the
instrument was installed, as well as to mitigate the impacts of low atmospheric pressures on the pump's
operation, a low air sampling rate of 7 L min$^{-1}$ for the pump model and 0.75 L min$^{-1}$ (at standard pressure
and temperature) for model 2537B were applied, based on the previous studies (Swartzendruber et al.,
2009; Zhang et al., 2015a; Zhang et al., 2016; Lin et al., 2019). Air was drawn in from the atmosphere
into the Tekran instrument, and the Hg was divided into GOM, PBM, and GEM inside the instrument for
analysis. A complete measurement cycle takes two hours. During the first hour, GOM was enriched on a
KCL-coated annular denuder, PBM was enriched on a quartz fiber filter (QFF), and GEM was directly
enriched on the gold tube of the Tekran 2537B and measured directly by cold vapor atomic fluorescence
spectroscopy (CVAFS). The collected PBM and GOM were desorbed in succession to Hg(0) at
temperatures of 800 °C and 500 °C in the following hour, respectively. Then the Hg(0) was measured by
Tekran 2537B. To ensure high data quality, the Tekran 2537B analyzer was set to use the internal Hg
source for automatic calibration every 23 h. The instrument was calibrated using an external Hg source
at the beginning and end of the monitoring period. The Tekran ambient Hg analyzer has been described
in detail in previous studies (Landis et al., 2002; Rutter et al., 2008; de Foy et al., 2016; Lin et al., 2019).
The monitoring data were also modified using the method from Slemr et al. (2016) as previous studies
suggested that there may be a low bias for low sampling loads (Slemr et al., 2016; Ambrose, 2017).
**2.3 Passive sampling of GEM concentration**
Passive samplers were set up at the same station during and after the active monitoring period to
better reflect the long-term pattern of local GEM concentration changes from April 4, 2019 to March 31,
2020. Sulfur-impregnated carbon (Calgon Carbon Corporation) was used as the sorbent for GEM (Guo
et al., 2014; Zhang et al., 2012; Tong et al., 2016; Lin et al., 2017). Passive samplers were deployed in
triplicate near the Tekran instrument at a height of ~2 m above the ground, and generally the passive
samplers were replaced three times per month (Table S1). After sampling, all samplers were sealed in a
three-layer zip-lock bag and transported to the laboratory, where they were then measured with the DMA-
80 (Milestone Inc., Itália). DMA-80 is an instrument that was used in accordance with US EPA Method
7473, using a combined sequence of thermal decomposition, mercury amalgamation and atomic
absorption spectrophotometry (Zhang et al., 2012). Hg concentrations in the atmosphere are then
calculated from the mass of sorbed Hg according to the equation obtained from our previous work (Guo
et al., 2014). The passive sampling method has been successfully applied to the Tibetan Plateau (Guo et
al., 2014; Tong et al., 2016) and North China (Zhang et al., 2012) in past studies. The use and quality
control of the Hg passive sampler have been described in detail in our previous studies (Zhang et al.,
2012; Guo et al., 2014; Lin et al., 2017). Similar passive sampling methods for Hg have been widely
used worldwide (McLagan et al., 2018).
**2.4 Meteorological data and other pollutants data**
During the monitoring period, the local temperature (with a precision of 0.1 °C), relative humidity
(with a precision of 1%), wind speed (with a precision of 0.1 m s$^{-1}$), wind direction (with a precision of
1°), air pressure (with a precision of 0.1 hPa), solar radiation (with a precision of 1 W m$^{-2}$), and UV index
(with a precision of 0.1 MEDs) were recorded at a 5-minute resolution by the Vantage Pro2 weather
station (Davis Instruments, USA).
Hourly measurement data of $PM_{2.5}$, $PM_{10}$, $SO_2$, $NO_2$, $O_3$, and CO concentrations and AQI index
were obtained from a nearby monitoring station in Nyingchi, which was hosted by the China Ministry of
Ecology and Environment and published by the China National Environmental Monitoring Center. The
measurements were conducted following the technical regulations for the selection of ambient air quality
monitoring stations (National Environmental Protection Standards HJ 664-2013) (Yin et al., 2019).
**2.5 Backward trajectory simulation**
To better understand the source of atmospheric GEM, the Hybrid Single-Particle Lagrangian
Integrated Trajectory (HYSPLIT) model was applied to calculate the backward trajectory many
atmospheric particles (Stein et al., 2015; Chai et al., 2017; Chai et al., 2016; Hurst and Davis, 2017; Lin
et al., 2019). HYSPLIT was developed by the US National Oceanic and Atmospheric Administration
(NOAA) and is a known tool for explaining atmospherically transported, dispersed, and deposited of
particles. The HYSPLIT model (https://www.arl.noaa.gov/hysplit/hysplit/) is a hybrid method that
combines the Lagrangian and Euler approaches. The Lagrangian method calculates the movement of air
parcels under the action of advection and diffusion, and the Euler method uses a fixed three-dimensional
grid to calculate the pollutant concentration. The backward trajectory simulation used Global Data
Assimilation System (GDAS) data with 1°x1° latitude and longitude horizontal spatial resolution and 23
vertical levels at 6 h intervals. The trajectory arrival height was set to 200 m a.g.l., which is about half
of the boundary layer height. We examined the effects of arrival height on the trajectories using
different arrival heights (20m, 50m, 200m and 500m respectively) in June 2019. The results show
that the calculated trajectories of the air masses are almost the same when the arrival height is below
500m (Figure S3). Each backward trajectory was simulated for 120 hours at 3 hours intervals for GEM,
which can cover China, Nepal, India, Pakistan, and the majority of western Asia. Cluster analysis was
performed after the trajectory calculation. Cluster analysis can help identify the average air masses
transport path by averaging similar or identical paths in the existing air masses paths, and provide
major directions of GEM transported to the measurement site.
**2.6 Principal components analyses**
Principal component analysis (PCA) is a data reduction method that can group some measured
variables into a few factors that can represent the behavior of the whole dataset (Jackson, 2005). PCA
has been employed in many previous Hg studies to analyze the relationships between Hg and multiple
pollutants and meteorological variables (Brooks et al., 2010; Cheng et al., 2012; Liu et al., 2007; Zhou
et al., 2019). All variables were normalized by standard deviation prior to running the PCA. To ensure
that the PCA is a suitable method for the data set in this study, the Kaiser-Meyer-Olkin measure of
sampling adequacy (> 0.5) and Bartlett's test of sphericity (p < 0.05) tests were performed in the initial
PCA run. Total variance and scree plots after rotation were used in the PCA analysis to determine the
factor numbers. Components with variance ≥1.0 were retained. Variables with high factor loadings
(generally > 0.5) were used to interpret the potential Hg source.
**3. Results and discussion**
**3.1 Hg species concentrations in Nyingchi**
During the whole monitoring period, the GEM, GOM and PBM concentrations at SET station were
$1.01\pm0.27$ ng m$^{-3}$, $12.8\pm13.3$ pg m$^{-3}$, and $9.3\pm5.9$ pg m$^{-3}$(mean±SD), respectively. Figure 2 shows the
GEM, GOM, and PBM concentrations and rainfall over the sampling period. Table S2 summarizes the
statistical metrics of Hg species, meteorological factors, and other pollutants in every monitoring period.
To further discuss the patterns of Hg concentrations, the entire monitoring period was divided into the
PISM period (before May 1) and the ISM period. The ISM period was further subdivided into three
periods (ISM1 – ISM3) according to changes in precipitation. The atmospheric Hg concentrations during
the PISM period ($1.20\pm0.35$ ng m$^{-3}$, $13.5\pm7.3$ pg m$^{-3}$, and $11.4\pm4.8$ pg m$^{-3}$, for GEM, GOM and PBM
respectively) were higher than those during the ISM period ($0.95\pm0.21$ ng m$^{-3}$, $12.7\pm14.3$ pg m$^{-3}$, and
$8.8\pm6.0$ pg m$^{-3}$, for GEM, GOM and PBM respectively). From ISM1 to ISM3, the average GEM
concentrations increased from $0.92\pm0.23$ ng m$^{-3}$, $0.92\pm0.18$ ng m$^{-3}$ to $1.04\pm0.21$ ng m$^{-3}$, while GOM
concentrations decreased sharply from $18.2\pm29.2$ pg m$^{-3}$, $13.5\pm5.5$ pg m$^{-3}$ to $6.0\pm5.0$ pg m$^{-3}$, PBM
concentrations decreased sharply from $15.4\pm7.9$ pg m$^{-3}$, $7.9\pm3.4$ pg m$^{-3}$ to $3.9\pm3.6$ pg m$^{-3}$. During the
PISM period, the GEM concentrations decreased continuously as the Indian monsoon developed and
intensified (Figure 2), which may indicate a change in the local GEM source as the wind field changes
from westerly to Indian monsoon. GEM concentrations remained relatively stable during ISM1 and ISM2
($0.92\pm0.23$ to $0.92\pm0.18$ ng m$^{-3}$), which may indicate that the source of GEM was relatively stable during
this period. However, at the end of the monsoon (ISM3), the GEM concentration started to increase
gradually to $1.04\pm0.21$ ng m$^{-3}$. There was no significant correlation between GEM concentration and
precipitation during the ISM period, which may be due to the stable chemical properties of GEM because
the air mass sources are relatively stable during the ISM period (Selin, 2009), while GOM and PBM
concentrations are strongly influenced by precipitation (Figure 2). With the increase in rainfall from
113.75 mm during ISM1 period to 373.28 mm during ISM2 period (total precipitation), the
concentrations of GOM and PBM decreased sharply from $18.2\pm29.2$ pg m$^{-3}$ and $15.4\pm7.9$ pg m$^{-3}$ to
$13.5\pm5.5$ pg m$^{-3}$ and $7.9\pm3.4$ pg m$^{-3}$, respectively. The considerable precipitation increase may be
responsible for the rapidly reduced GOM and PBM concentrations, as they are easily deposited in the
atmosphere with precipitation (Lindberg and Stratton, 1998; Seigneur et al., 2006). GOM and PBM
concentrations continued to decline from ISM2 to ISM3, however, the trend in precipitation was reversed.
This may indicate that less GOM and PBM were transported to the SET station or with fewer local
sources during ISM3. In a previous study, Huang et al. (2015) found that even with heavy rain during the
monsoon period, the total Hg concentration in precipitation in the SET region was small but still
considerable, suggesting that there may be a stable source of Hg in the SET region during the ISM period.
The high total Hg concentration in precipitation during ISM may indicate that local emissions could be
important sources during ISM period.
Figure 3 shows the results of the GEM concentrations obtained through passive samplers throughout
the year. The average GEM concentration is 1.12±0.28 ng m$^{-3}$, which is slightly higher than the average
GEM concentration during the Tekran monitoring period (1.01±0.27 ng m$^{-3}$). **In terms of seasonal**
**variation, average GEM concentrations were the lowest in summer (1.03±0.09 ng m$^{-3}$), with almost**
**identical average concentrations in spring, autumn and winter (1.14±0.28 ng m$^{-3}$, 1.16±0.35 ng m$^{-3}$**
**and 1.14±0.28 ng m$^{-3}$, respectively). This is different from the trends of GEM concentrations in the**
**surrounding areas, where the highest GEM concentrations in Nam co, Mt. Ailao, Mt. Waliguan**
**and Mt. Gongga (Yin et al., 2018; Zhang et al., 2016; Fu et al., 2012; Fu et al., 2008) were all seen**
**in summer, which may indicate that the Indian summer winds that bring high GEM concentrations**
**to these areas do not present similar effect on the SET region.** For the variation throughout the year,
the GEM concentration in May and June is the lowest with an average concentration of only 0.97±0.18
ng m$^{-3}$, while November and December have the highest GEM concentrations (1.24±0.37 ng m$^{-3}$). The
average GEM concentration is lower (1.02±0.09 ng m$^{-3}$) during the ISM period (from May to August)
and higher during the westerly circulation period (1.16±0.32 ng m$^{-3}$); however, the GEM concentration
during westerly circulation period has large fluctuations. Since there are almost no local industries and
less human activity in Nyingchi, this difference may indicate a higher input of pollutants introduced by
westerly circulation.
Table 1 summarizes the GEM, GOM, and PBM concentrations from research papers of high-altitude
regions around the world. Compared to other high-altitude sites, the GEM concentrations in the SET
region were relatively low and did not reach the average GEM concentration level in the Northern
Hemisphere (~ 1.5-1.7 ng m$^{-3}$). Compared to previous studies of high elevation (> 2000 m a.s.l) regions,
only Concordia Station in Antarctica had lower GEM concentrations than those observed at the SET
station. Ev-K2, Nam Co, Qomolangma, and Shangri-La, the nearest monitoring stations to the SET
station and at higher altitudes, had higher GEM concentrations than those at the SET station. In particular,
the GEM concentration at Shangri-La was more than two-fold of that at the SET station. The differences
in the GEM concentrations among them may be mainly due to their different climatic conditions and
different monsoon control zones, which result in different pollutant source regions and air mass transport
trajectories. The Shangri-La station may be influenced by anthropogenic emissions within and outside
China, and therefore, has higher GEM concentrations. For Ev-K2 and Qomolangma stations, which are
under the influence of the ISM, they may be directly exposed to air masses with high concentrations of
pollutants transported from India and Nepal. Although there are extreme deposition processes during the
climbing process to both Ev-K2 and Qomolangma stations, some Hg may survive reach the stations (Lin
et al., 2019). The GOM concentrations at the SET station were approximately at the average level among
the monitored sites. PBM concentrations were relatively low at the SET station, which may be due to the
high rainfall in the YZB Grand Canyon, easily washing away particulate Hg by rainwater.
The lower GEM concentrations during the ISM period may indicate that the pollutant sources of the
SET region changed with the weakening of the westerly circulation and the strengthening of the Indian
monsoon. Previous studies (Lin et al., 2019; Gong et al., 2019a; Wang et al., 2015a) indicated that
pollutants from the heavily polluted Indian subcontinent may be transported to the Tibetan Plateau under
the action of ISM, resulting in increased local pollutant concentrations on the plateau. This was verified
at the Qomolangma, Nam Co, and Mt. Ailao stations, where GEM concentrations were higher during the
ISM period than the PISM period (Lin et al., 2019; Yin et al., 2018; Zhang et al., 2016). However, in our
study, the SET station observed lower Hg species concentrations during the ISM than the PISM period.
For GEM, the decrease in concentration may be due to the absorption effect from the dense vegetation
during the monsoon period (Fu et al., 2016b), while air masses from the Indian Ocean bring large amounts
of halogens (Fiehn et al., 2017), which may react with and deplete GEM. For GOM and PBM, increased
concentrations were observed during the ISM1 period, whereas their concentrations decreased sharply
during the ISM2 and ISM3 periods. The decreases in GOM and PBM concentrations may be mainly due
to the rapid increase in local precipitation during the Indian monsoon period, which starts after the
monsoon enters China from northwestern India. A large amount of water vapor from the Indian monsoon
climbs more than 3,000 m within ~100 km in the YZB Grand Canyon, producing considerable
precipitation. Therefore, GOM and PBM may deposit during transportation and are unable to reach the
Nyingchi area.
Table S3 shows the variations of Hg species, meteorological factors and other pollutants from
June 1 to 4, 2019. High GOM concentrations were observed on June 2 and 3, and very high solar
radiation and UV Index were also observed in these days. PBM concentrations, relative humidity
and $O_3$ were low during this period. The solar radiation was nearly twice the mean value of the
ISM1 phase (162.79 W m$^{-2}$, Table S2), and thus higher solar radiation might contribute to the higher
GOM concentrations. PBM might be partly converted to GOM, but the decrease in PBM
concentration was less than the increase in GOM concentration. Generally, high $O_3$ concentrations
should be observed at higher solar radiation (Kondratyev et al., 1996), but low $O_3$ concentrations
were found at Nyingchi, suggesting that $O_3$ may contribute to the formation of GOM. The oxidation
of GEM by OH and $O_3$ to generate GOM has been discussed in previous studies with model
simulation (Sillman et al., 2007), which may explain the reduced concentration of $O_3$, while OH
radicals may be associated with high solar radiation. The mechanism of GOM formation should be
further explored in future studies.
**3.2 Diurnal Variation**
Figure 4 shows the diurnal variation of Hg species and the concentrations of other pollutants during
the entire monitoring period. In general, the Hg species concentrations varied significantly during the
PISM period, and the diurnal variation was relatively small after entering the ISM period. During the
PISM period, the GEM concentrations were relatively low during the daytime (average 1.07 ng m$^{-3}$ from
11:00 to 18:00), gradually accumulated after sunset, and finally reached a relatively stable high value
(average 1.26 ng m$^{-3}$) at night. During the ISM period, the GEM concentration variation pattern was not
as pronounced as during the PISM period, with the lowest GEM concentration of the day usually
occurring around sunrise (0.83, 0.80, 0.88 ng m$^{-3}$ for ISM1-3, respectively). During ISM1, the GEM
concentration reached a high value around 9:00 a.m., fluctuated less during the daytime, reached a
maximum value in the evening, and gradually dissipated in the early morning. During ISM2, the
maximum value was reached at approximately 16:00, was more stable in the evening, and gradually
dissipated in the early morning. During ISM3, the maximum value was reached at approximately 20:00
and dissipated in the early morning. The average of the daily maximum values were 1.04, 1.00, 1.16 ng
m$^{-3}$ for ISM1-3 periods, respectively. After midnight, GEM concentrations gradually decreased. In
general, the daily variation of GEM in previous research were about 0.2-0.9 ng m$^{-3}$ globally (Fu et al.,
2012a; Fu et al., 2008; Fu et al., 2010; Lin et al., 2019; Zhang et al., 2015a), and were lower at the SET
site (0.21, 0.20, 0.28 for ISM1-3 periods, respectively). For GOM and PBM, the diurnal variations
showed U-shaped variation patterns during the PISM period. During this period, the concentrations of

GOM and PBM reached low values between 10:00 and 14:00, then gradually accumulated and peaked around midnight. After midnight, the concentration gradually decreased to its lowest point. During the ISM1 period, GOM and PBM concentrations were higher in the afternoon and evening, and showed a decreasing trend after midnight. During ISM2-3, GOM and PBM did not show clear daily variation patterns. Except for the ISM2 period, there was little difference between GOM and PBM concentrations during the other periods, which may be due to similar sources and behavioral patterns in the environment. In contrast, during the ISM2 period, more precipitation (Figure 2) led to a sharp decrease in PBM concentrations, and it is speculated that GOM may have additional sources during this period. The oxidation of GEM by OH and $O_3$ to generate GOM may be a possible reason for the high GOM concentration (Sillman et al., 2007). However, the mechanism of GOM formation should be further explored.

Compared with other Hg monitoring in previous studies, some diurnal variation trends of Hg at the SET site were unique. In previous studies (Sprovieri et al., 2016; Yin et al., 2018; Zhang et al., 2015a; Zhang et al., 2016; Fu et al., 2012a; Fu et al., 2010; Lan et al., 2012), a common pattern of highest concentration around noon and lowest concentration before sunrise was mostly observed. The decrease in GEM concentration at night may be due to the interaction of pollutants from regional emissions and long-range transport(Fu et al., 2008; Fu et al., 2010). After sunrise, partial GEM re-emission occurs in the sunlight, along with the mixing effect of the residual boundary layer downward, which may lead to an increase in GEM concentration (Mao and Talbot, 2012; Selin et al., 2007; Weiss-Penzias et al., 2009; Talbot et al., 2005). The height of the boundary layer increases after noon during the daytime, which produces dilution of GEM at the surface and may be the reason for the decrease in GEM concentration in the afternoon. The GEM diurnal variation pattern at the SET is particularly special during the PISM period, while a similar variation pattern was also observed at the Qomolangma site in our previous research (Lin et al., 2019), which is another high-altitude site with a sparse population and rare industry. This similar pattern suggests that they have a similar mechanism of GEM diurnal variation. Considering that neither site has an obvious local source of GEM, the variation in GEM concentrations may only be subject to these mechanisms. Similar to the study of Qomolangma, the variation in the boundary layer height may be one of the reasons for the diurnal variation of GEM concentration in the SET region. The stable and low height nocturnal boundary layer at night causes the GEM concentration to gradually

concentrate, and the boundary layer gradually increases to a higher altitude after sunrise. The gradual increase in GEM concentration during the daytime may be due to the reduction of GOM from nearby local snowy mountains (Lalonde et al., 2003; Lalonde et al., 2002) or long-range transported GEM brought in by airflow (Lin et al., 2019). During the ISM period, the nighttime GEM dissipation may be due to the fact that this area enters a rapid leaf-growing season (Fu et al., 2016b) after entering the ISM period, that the air masses from the Indian Ocean bring a large amount of halogens (Fiehn et al., 2017), and that depletion of GEM occurs under the boundary layer at night.

**3.3 Source identification for atmospheric Hg in Nyingchi**

To further investigate the contributions of different sources to the SET site, air mass back trajectory simulation and trajectory cluster analyses were performed for GEM. Figure 5 shows the cluster analysis results for the PISM and ISM1-3 periods. Based on the results of the total spatial variation index, 3-5 clusters were grouped for each period. Each clustered trajectory contained detailed information about the trajectory from the source region to the SET site, the trajectory frequency during the period, and the concentrations of the pollutants carried by the air mass when the trajectory arrives.

During the PISM period (Figure 5a), the trajectories mainly originated from or passed through central India, northeastern India, and central Tibet, and moved along the southern border of the Himalayas Mountains. During this period, the meteorological factors at Nyingchi were mainly controlled by westerly circulation. The cluster with the highest concentration (cluster2, with GEM concentration of 1.19 ng m$^{-3}$) originated from or passed through central Tibet, accounting for 13.75% of all trajectories in this period. Although the GEM concentrations of the cluster were relatively high during this period, they were still lower than the background GEM concentration in the Northern Hemisphere (~ 1.5-1.7 ng m$^{-3}$), indicating that the air mass transported to the SET station is relatively clean. Cluster1, from the southern border of the Himalayas, was relatively high in proportion (with a frequency of 78.58%), mainly controlled by the southern branch of the westerly circulation, and has a relatively low concentration (1.12 m$^{-3}$). This cluster made a turn in the south of SET station and began to ascend toward the Tibetan Plateau. According to the UNEP reports, Hg emission intensities along the trajectory paths were weak (UNEP, 2018; UNEP, 2013b).

During the ISM period (Figure 5b-d), the trajectories of arrivals at the SET site changed significantly with the onset and rise of the Indian monsoon. The clusters undergo a slight counter-clockwise rotation.

As the source of the air mass changes and the monsoon enters the plateau, it is possible that the concentrations of pollutants decrease because of the change in the source region. With the development of the Indian monsoon, it brings an abundance of water vapor (Ping and Bo, 2018), which may cause strong deposition during transportation. During the ISM1 period (Figure 5b), both the rising monsoon and the tail of the westerly circulation control the meteorological factor at the region, causing the transported air masses to exhibit complex trajectories and combined effects. The cluster with the highest concentration (cluster4, 0.96 ng m$^{-3}$, and 14.02%) mainly came from or passed through central India. Cluster3 share almost the same transport path with cluster4 while having shorter length and lower GEM concentration, which may indicate that cluster4 was affected by GEM emission in central India. The trajectory with the largest proportion (cluster1, 43.94%) had a relatively short path, mainly from northeast India, and showed very low GEM concentration (0.92 ng m$^{-3}$). Based on the existing atmospheric Hg emission inventories (Simone et al., 2016; UNEP, 2018; UNEP, 2013b), the Hg emission intensities in cluster1 transport path are very low, which may be the reason for the low GEM concentration in this cluster.

During the ISM2 period (Figure 5c), a typical period of Indian monsoon, almost all trajectories came from or passed through the southern part of the SET site and were influenced by the monsoon. The GEM concentration of cluster trajectories at this stage was below 1.00 ng m$^{-3}$. The majority of trajectories (cluster2, 85.82%) through the YZB Grand Canyon to the SET station and have a short transport path, which may be related to the high resistance of the dense vegetation in summer. Only about 2.24% of the trajectories originated from central Tibet with very low GEM concentration (cluster3 with 0.99 ng m$^{-3}$). During this period, the ISM originated from the Indian Ocean brought a large amount of water vapor and caused considerable precipitation during the transportation. At the same time, the areas through which the trajectory passed were sparsely populated and underdeveloped and were unable replenish Hg species to the air masses. The range of GEM concentrations during the ISM2 phase was extremely small (Figure 2), which may indicate that under the strongly Indian monsoon, the main source region, transport path, and mechanism of transportation during this period remain stable.

During the ISM3 period (Figure 5d), the Indian monsoon remained controlling the meteorological factors at the SET station, but its intensity was weakened, and the precipitation in the Nyingchi area was greatly reduced. The trajectories transmission distances are all short. All of the trajectories still came

from south of SET station and transported through the YZB Grand Canyon. It is difficult to distinguish
these clusters, but according to the UNEP (2018) Report, it is clear that the areas for which the clusters
passed through have very little emission. The GEM concentration at SET increased compared with the
ISM1-2 periods (average at 0.92 ng m$^{-3}$ in ISM1 and ISM2, and 1.04 ng m$^{-3}$ in ISM3 period, respectively).
This may indicate that the GEM source is farther away. At the end of the ISM3 period, the GEM
concentration showed an upward trend (Figure 2), which may be due to the weakening of the influence
of the monsoon. A shortened trajectory at the end of the monsoon period was also observed in another
study at a nearby site (QNNP) (Lin et al., 2019), which may indicate the withdrawal of the monsoon.
We also calculated backward trajectories for the passive sampler monitoring period. Figure S4
shows the trajectories of air masses arriving at the SET station in different seasons. Due to the low
accuracy of the data obtained from passive sampling, we didn't combine the GEM concentrations from
the passive sampler monitoring with the trajectories here. Except for winter, the vast majority of
trajectories originated from the south of the SET station, and most of the trajectories are short in distance.
This may be related to the complex local topography, which may also suggest that long-distance transport
has limited effect on SET station. There is a partial shift of the backward trajectory from the southwest
to the south in spring, compared to summer, which may originate mainly from the influence of the Indian
monsoon. The abundance of precipitation, halogens from the Indian monsoon, and rapid growth of
vegetation during the monsoon period may have depleted Hg species, and resulted in the lower GEM
concentrations in summer. Trajectories from the northern branch of the westerly circulation were more
abundant in autumn compared to winter, but did not appear to have an impact on local mean GEM
concentrations. Because of the large concentration variations in the passive sampling monitoring, we
aggregated the trajectories for the periods of high concentrations (GEM concentrations above 1.5 ng m$^{-}$
$^{3}$) and low concentrations (GEM concentrations below 1.0 ng m$^{-3}$), and performed a cluster analysis. The
majority of trajectories in both categories were from the southern part of the SET station and were of
similar length (Figure S5), which indicates that the differences in concentrations monitored by passive
sampling may not be related to external transport.
**3.4 Hg concentration controlling factor indicated by PCA results in Nyingchi**
Overall, 4-5 factors were resolved for each period from the PISM to ISM3 periods. Some factors
are unique to each period, and certain factors are found throughout the monitoring period. Only Hg-
related components were reserved here and four underlying PCA factors are summarized (Table 2). They
were assigned as long-distance transport, local emissions, meteorological factor, and snow melt factor.
The long-distance transmission factor (F1) found in the PISM and ISM3 periods mainly contain
GEM, wind speed, CO (positive loading), temperature, and $SO_2$ (negative). GEM could be considered
an indicator of long-distance transportation due to its long lifetime in the atmosphere, especially when
GOM and PBM are not significant in this factor. This factor indicates that the long-distance transportation
of GEM may mainly occurs in the pre-monsoon and the end of the monsoon period, which is similar to
the trajectory analysis in Section 3.3. The negative correlation between GEM and temperature may
indicate that the long-distance transport of GEM during the PISM period occurs mainly during periods
of lower temperatures. Compared with the diurnal variation of GEM during the ISM period (Figure 4),
it is possible that the increasing GEM concentration in the evening in the PISM period is mainly due to
the long-distance transportation of GEM.
Factor 2 involved GOM and PBM (high positive loading) in each period, mainly with positive $O_3$,
$PM_{10}$, $PM_{2.5}$, and negative temperature. GOM concentrations were positively correlated with PBM
concentrations, which implies that these two species probably originated from the same sources. The
high positive loadings of PBM, GOM, and some particle pollutants may indicate that the main source of
PBM and GOM is local emissions. The long-distance transport of particle pollutants from the Indian
subcontinent may have heavy wet deposition when the air mass climbs into the Tibetan Plateau and
cannot reach Nyingchi successfully. Thus, the local monitored particle pollutants, as well as easy-
deposition pollutants, may mainly originate from regional emissions. One possible source is from yak
dung; in the Tibetan Plateau, yak dung is a widely used household biofuel (Xiao et al., 2015) and the
burning of yak dung may release Hg and other particulate matter (Rhode et al., 2007; Xiao et al., 2015;
Chen et al., 2015).
Meteorology factor (F3) was found during the ISM period with positive temperature, wind speed,
solar radiation, and negative humidity and rain, which are likely associated with meteorological
conditions. This factor shows that meteorological conditions may profoundly affect the overall local
pollutant distributions during the ISM period, which suggests that the air mass carried by the ISM not
only cannot increase the long-distance transportation of pollutants to the Nyingchi area, but may also
reduce the local contribution of pollution. For existing pollutants, the strong positive loading of solar

radiation may indicate that pollutant reactions under strong radiation are relatively active in this high-altitude region. The strong negative humidity and rain may indicate that rain has played a strong role in the cleaning process, especially during the ISM1 and ISM2 periods, when precipitation is relatively strong.

Factor 4 had a strong positive correlation with GEM, ROM, and solar radiation, and negative loading with humidity during the ISM1 period. This suggests that as solar radiation increases in the afternoon, more GEM and GOM are emitted to the air. The influence of increasing solar radiation may reflect the snow/ice melt process. which have been proved to be able to increase atmospheric GEM concentration (Huang et al., 2010; Dommergue et al., 2003). GEM may originate from the evaporation of snow melting and/or be driven by the photoreduction of snow $Hg^{II}$ (Song et al., 2018). The simulation indicated that the oxidation of GEM may occur at the snow/ice interface in the action of solar radiation, and may lead to extra GOM release. The peak concentrations of GEM and GOM both appeared in the afternoon during the ISM1 period, when the solar radiation was the highest and humidity was the lowest. The increase in GEM and GOM concentrations may be related to solar radiation, according to the PCA results.

The PCA results provide some new insights into the sources of Hg species. During active monitoring period, long-distance transport of GEM was the main source of SET station and only occurred at PISM and ISM3. Given the low GEM concentrations in ISM1 and ISM2, it is reasonable that PISM and ISM3 are the main long-distance transport periods for GEM. For GOM and PBM, on the other hand, local sources appear to be more important during active monitoring period. This may be related to the fact that GOM and PBM deposit more easily and have complex transport paths to the SET station. The local sources of GOM and PBM are inconclusive. The concentrations of GOM and PBM monitored at the SET station are not high and the local emissions can be assumed to be small. They might come from yak dung burning or other local sources by the local residents (Rhode et al., 2007; Xiao et al., 2015; Chen et al., 2015), and/or the strong solar radiation and snow surface reaction, which needs to be confirmed by further field experimental studies.

**3.5 Implications**

The Tibetan Plateau is a direct invasion target of the ISM. Blocked by the high altitude of the Himalayas, the Indian monsoon could bypass the high mountains and enter Tibet via the YZB Grand

Canyon. When the summer monsoon enters Tibet, pollutants from India and the Indian Ocean, as well as
large amounts of water vapor, may be carried along with the air masses (Lin et al., 2019; Yang et al.,
2013; Wang et al., 2018). Located in the water vapor channel where the Indian monsoon enters, Nyingchi
is believed to receive a large amount of foreign air masses (Yang et al., 2013). Considering that Nyingchi
has little local emission because of the sparse population and lack of industry, the pollutants present in
the area should mostly have been transported by monsoons over long distances. However, our monitoring
results show that during the ISM period, the GEM concentrations in the Nyingchi are extremely low
(0.95±0.21 ng m$^{-3}$); lower than the background GEM concentration in the Northern Hemisphere and the
GEM concentrations observed at surrounding monitoring sites in the literature (Table 1).

561        The low concentration during the ISM period may be related to the regional deposition process and

complex regional terrain. When monsoon winds carry large amounts of Indian Ocean moisture and enter
the YZB Grand Canyon, strong wet deposition occurs during transport due to an increase in elevation
and a decrease in temperature. The process of rainwater scouring from wet deposition may result in
significant deposition of pollutants from carried air masses (Lindberg and Stratton, 1998; Seigneur et al.,
2006). Meanwhile, the air flow in the canyon is slow owing to the complex terrain. The slow migration
of the air mass further strengthens the deposition process. In addition, during the ISM period, the dense
forest in the canyon may deplete some of the Hg during transport (Fu et al., 2016b). Therefore, pollutants
from the Indian subcontinent struggle to go deep into the Tibetan Plateau during the ISM period. The
deposited pollutants may flow into the downstream area via rivers to Southeast Asia and South Asia.
Additional wet deposition monitoring along the YZB Grand Canyon in the future may provide more
evidences on transportation mechanisms. However, long-distance transboundary transport remains an
important mechanism of GEM distribution in this area during the period of westerly circulation. As
discussed in Section 3.1, the GEM concentration in Nyingchi during PISM period (1.20±0.35 ng m$^{-3}$)
was much higher than that during the ISM period (0.95±0.27 ng m$^{-3}$). The high GEM concentration
during the PISM period may indicate that a large amount of external Hg entered the Nyingchi area during
the non-ISM period, and thus monitoring of isotopic atmospheric Hg in future studies or accurate model
simulations are needed to provide better evidences.

579        The results of our previous study on Qomolangma were different from those in Nyingchi.

Qomolangma site locates on the northern side of the Himalayas, a typical terrain on the southern edge of
the Tibetan Plateau. The Nyingchi site locates in a typical pathway for air masses to enter the Tibetan
Plateau. Both sites locate in sparsely populated areas, far from human activity, making them ideal clean
locations to study the behavior of Hg species. Hg species monitoring in both sides could help explain the
possible transboundary transport patterns. In terms of the concentration distributions of Hg species, both
sites showed low concentrations, with slightly higher GEM concentrations identified at Qomolangma
site. The diurnal variations in the concentrations of Hg species are unique in both areas, as there are
relatively little anthropogenic disturbances, but Nyingchi is surrounded by greater elevation variation
and more complex terrain, and thus the diurnal variation is subject to more natural disturbance factors.
In terms of Hg species from long-range transport, Qomolangma was mainly affected by monsoonal
transport from India during the ISM period, showing the increases in the concentrations of GEM.
Nyingchi, on the contrary, has low GEM concentrations during the ISM. Although receiving almost the
same monsoonal influences from India, the intensity of the transport and the subsidence on the transport
path may be responsible for the large differences in the concentrations of Hg species and their
environmental behavior between the two sites. Together, they represent two typical transboundary
transport patterns of Hg in the Tibetan Plateau.
**4. Conclusions**

597         Comprehensive Hg species monitoring was carried out in Nyingchi, a high-altitude site in the

southeast of the Tibetan Plateau. Nyingchi is located on the main pathway for water vapor carried by the
monsoon to enter the Tibet Plateau during the ISM period, which could characterize the spread of
pollutants from the Indian subcontinent. The concentrations of GEM and PBM during the PISM period
were significantly higher than those during the ISM period, and the concentration of GOM during the
PISM period was relatively higher than that during the ISM period. Data from passive sampler
monitoring showed that, average GEM concentrations were the lowest in summer, with almost identical
average concentrations in spring, autumn and winter. The concentrations of Hg species in Nyingchi is
particularly low, compared with other high-altitude stations around the world. GEM concentration shows
a distinct and unique diurnal variation, with a gradual increase in GEM concentration during the day and
a maximum concentration at night. This diurnal variation may be due to the re-emission of GEM by
snowmelt and the trapping effects of pollutants by the very low planetary boundary layer at night.

609         According to the trajectory model, the trajectories of arrivals changed significantly with the onset

and rise of ISM. Except for winter, the vast majority of trajectories originated from the south of the SET station, and most of the trajectories are short in distance. Through comprehensive PCA analysis using local meteorological conditions and multiple pollutants, long-distance transport, local emissions, meteorological factor, and snowmelt factor have been identified to affect local Hg species concentrations. PCA analysis results also indicate that local emission contributes between PISM and ISM3, while the long-distance transportation plays a role during PISM and ISM3. The deposition condition and vegetation distribution in the YZB Grand Canyon have significant influences on the transport of Hg species. The Grand Canyon on the one hand reduces atmospheric Hg species concentrations in Nyingchi, but at the same time poses some risks of high Hg species concentrations downstream. Our work reveals the effect of the YZB Grand Canyon on atmospheric Hg transport, while the pathways associated with the deposition of GOM and PBM, and the destinations of GEM should be studies in more detail in the future.

**Acknowledgments**

This study was funded by the National Natural Science Foundation of China (Grant #41630748, 41977311, 41977324, 41821005). The authors are grateful to NOAA for providing the HYSPLIT model and GFS meteorological files. We also thank the staffs of the South-East Tibetan Plateau Station for Integrated Observation and Research of Alpine Environment, Chinese Academy of Sciences on Nyingchi for field sampling assistance.

Data availability. All the data presented in this paper can be made available for scientific purposes upon request to the corresponding authors.

Author contributions. HL,XW, YT, QZ and XY designed the research and performed field measurements. HL YT and CY performed the data analysis and model simulations. HL lead the paper writing. LC,SK,LL,JS and BF contributed to the scientific discussion and the paper preparation.

Competing interests. The authors declare that they have no conflict of interest.

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

Table 1. Comparison of atmospheric Hg concentrations at high elevation (>2000m a.s.l) stations

| Site | Country | Lat & Lon | Elevation | Type | Time Period | GEM or TGM mean±SD, ng/m3 | GOM mean±SD, pg/m3 | PBM mean±SD, pg/m3 | Reference |
|---|---|---|---|---|---|---|---|---|---|
| Concordia Station | Antarctica | -79.1/123.35 | 3220 | Remote | 2013-2014 | 0.80±0.25 | - | - | Sprovieri et al., 2016 |
| **SET** | **China** | **29.77/94.74** | **3263** | **Remote** | **2019** | **1.01±0.27** | **12.8±13.3** | **9.3±5.9** | **This study** |
| Ev-K2 | Nepal | 27.96/86.81 | 5050 | Remote | 2012-2014 | 1.19±0.30 | - | - | Sprovieri et al., 2016 |
| Nam Co | China | 30.78/90.99 | 4730 | Remote | 2012-2014 | 1.33 ±0.24 | - | - | Yin et al., 2018 |
| Qomolangma | China | 28.37/86.95 | 4276 | Remote | 2016 | 1.42±0.37 | 21.4±13.4 | 25.6±19.1 | Lin et al.,2016 |
| Kodaicanal | India | 10.23/77.47 | 2333 | Rural | 2013-2014 | 1.54±0.23 | - | - | Sprovieri et al., 2016 |
| Col Margherita | Italy | 46.37/11.79 | 2545 | Rural | 2014 | 1.69±0.29 | - | - | Sprovieri et al., 2016 |
| Lulin | China | 23.51/120.92 | 2862 | Remote | 2006-2007 | 1.73±0.61 | 12.1±20.0 | 2.3±3.9 | Sheu et al., 2010 |
| Mt. Walinguan | China | 36.29/100.90 | 3816 | Remote | 2007-2008 | 1.98±0.98 | 7.4±4.8 | 19.4±18.0 | Fu et al., 2012c |
| Mt.Ailao | China | 24.53/101.02 | 2450 | Remote | 2011-2012 | 2.09±0.63 | 2.2±2.3 | 31.3±28.0 | Zhang et al., 2016 |
| Shangri-La | China | 28.02/99.73 | 3580 | Rural | 2009-2010 | 2.55±0.73 | 8.2±7.9 | 38.8±31.3 | Zhang et al., 2015 |
| Mt. Leigong | China | 26.39/108.20 | 2178 | Remote | 2008-2009 | 2.8±1.51 | - | - | Fu et al., 2010b |


Table 2. PCA Factor Loadings (Varimax Rotated Factor Matrix) for Hg in Nyingchi, Tibet, China

| tentative identification | | GEM | PBM | GOM | Temp | Hum | Wind Speed | Rain | Solar Rad. | CO | NO$_2$ | O$_3$ | PM$_{10}$ | PM$_{2.5}$ | SO$_2$ | Variance Explained |
|---|---|---|---|---|---|---|---|---|---|---|---|---|---|---|---|---|
| long distance | PISM | **0.92** | | 0.10 | **-0.79** | | **0.64** | -0.15 | | 0.28 | 0.43 | | | -0.27 | **-0.73** | 19.86 |
| transport | ISM3 | **0.78** | | | -0.22 | 0.26 | 0.18 | 0.49 | -0.13 | **0.76** | | 0.29 | -0.11 | | **-0.83** | 17.05 |
| local emission | PISM | 0.13 | **0.91** | **0.92** | | | -0.20 | | | 0.22 | -0.47 | -0.13 | 0.12 | | -0.44 | 15.96 |
| | ISM1 | 0.26 | **0.56** | 0.19 | | | | -0.12 | -0.16 | | -0.12 | | **0.69** | **0.86** | 0.32 | 12.97 |
| | ISM1 | 0.17 | **0.60** | 0.16 | -0.40 | | 0.19 | -0.14 | -0.11 | | | **0.91** | 0.22 | -0.19 | | 11.11 |
| | ISM2 | **0.50** | **0.89** | **0.77** | **-0.51** | **-0.52** | **0.71** | -0.12 | 0.10 | 0.14 | | **0.71** | **0.71** | **0.72** | | 30.26 |
| | ISM3 | 0.25 | **0.96** | **0.95** | -0.13 | -0.19 | -0.12 | -0.34 | 0.27 | | | -0.15 | | 0.16 | 0.32 | 16.46 |
| meteorology | ISM1 | | -0.18 | 0.11 | **0.82** | **-0.77** | **0.62** | **-0.80** | **0.68** | | **-0.59** | | 0.13 | | 0.31 | 23.46 |
| | ISM2 | **-0.51** | | -0.23 | **0.66** | **-0.80** | 0.21 | **-0.79** | **0.85** | | -0.10 | 0.25 | 0.43 | 0.18 | -0.13 | 22.02 |
| | ISM3 | -0.19 | 0.17 | | **0.85** | **-0.89** | | -0.30 | **0.88** | -0.44 | | 0.46 | 0.31 | 0.21 | | 21.49 |
| melt | ISM1 | **0.78** | 0.13 | **0.85** | | **-0.57** | | | **0.50** | | 0.33 | 0.24 | 0.21 | 0.21 | 0.25 | 15.94 |

Note: Variables with high factor loadings (> 0.5) were marked in bold. For readability, variables with very low factor loadings (<0.1) are not presented.

Figure Captions:

Figure 1. Location of the South-East Tibetan Plateau Station for Integrated Observation and
Research of Alpine Environment (SET station or Nyingchi station, the red star). SET station is
located in a water vapor channel from the Ganges River Plain to the Tibetan Plateau. The red dot is
Lhasa, the capital city of the Tibet Autonomous Region, which is the most densely populated city
in Tibet; the other red dot is the nearest town to the monitoring site, Bayi Town.
Figure 2. Time serious of GEM, GOM, and PBM concentrations and the rainfall over the sampling
period. The GEM concentration resolution is 5 min, and the GOM, PBM, and rain resolutions are 2
hours. According to the characters of monsoon development and precipitation, the monitoring
periods are divided into four segments, namely PISM (before May), ISM1 (1 May- 2 June), ISM2
(3 June – 8 August), and ISM3 (after 9 August).
Figure 3. GEM concentrations obtained through passive samplers throughout the year. The black
squares represent the atmospheric Hg concentrations obtained by passive sampling, and the upper
and lower error lines are the standard errors of the passive samples monitored during the same time
period. The red dots represent the GEM concentrations obtained through the Tekran instrument. The
green horizontal line indicates the average of the atmospheric mercury concentrations during this
period.
Figure 4. Diurnal variation of Hg species, concentrations of some other pollutants and
meteorological information from PISM to ISM1-3 periods. The short horizontal line represents the
concentration error range for each time period.
Figure 5. Clusters of the back trajectory analysis from SET site during PISM to ISM3 periods. The
thickness of the line represents the ratio of the cluster in the time period, the background is the
globally Hg emission inventory developed by UNEP(UNEP, 2013a).
