# Peer review of "Water Vapor Channel in the Southeast Tibetan Plateau"

_Atmospheric Chemistry and Physics, 2021_

## Author Response (AR1)

**Responses to the Reviewers' Comments**

 **First Observation of Mercury Species on an Important Water Vapor Channel in the**

 **Southeast Tibetan Plateau**

Dear editor and reviewer,

We greatly appreciate the useful comments and suggestions from the editor and reviewers. We think the novelty and importance of this study have been acknowledged by the reviewers. We have revised the manuscript thoroughly based on the reviewers' comments. Detailed point by point responses are provided below. All the revisions have been highlighted in blue in the revised manuscript. We hope the revised manuscript could meet the standard of ACP. Thanks again for your consideration.

**Anonymous Referee #2**

The manuscript entitled 'First Observation of Mercury Species on an Important 2 Water Vapor Channel in the Southeast Tibetan Plateau' by Line et al. presents ~5 months of speciated mercury concentrations (using online and offline sampling) at Nyingchi during the period preceding and during the Indian Summer Monsoon (ISM). This site is located in an important water vapor channel and thus is ideal for investigating the transport of pollution to the Tibetan Plateau. The authors divide the ISM into three periods, then use back trajectory clustering analysis and principal component analysis to investigate the sources and source regions affecting mercury concentrations. The authors found the PISM periods to be affected by westerly circulation with higher levels of GEM, a distinct diurnal pattern, with long-range transport and local emissions being important factors. While the ISM period was affected by transport from the Bay of Bengal and the Indian Ocean, with lower levels of all mercury species, a different diurnal pattern compared to PISM, and local emissions, meteorology, and snowmelt. They concluded wet deposition and uptake by vegetation to be responsible for the low concentrations observed during the ISM. This manuscript presents the first results from this location and coupled with their previous study from Qomolangma Natural Nature Preserve present an important analysis of pollution entering the Tibetan Plateau. However, there are points where the manuscript could be improved. Their interpretation is sound although requires more discussion. While the manuscript is readable, there are improvements to the language that would aid in the readability. Overall, I recommend the publication of this manuscript after addressing the major revisions outlined below.

**Response:**

Thanks for your detailed comments and suggestions. We have polished the language of the manuscript, updated the cited references, extended the discussion and revised the figure location accordingly. Please see the revised manuscript. All the revisions have been highlighted in blue. Detailed responses to your comments are provided as follows.

**General Comments**

**Comment #1**

It is important to make a distinction between which species of mercury the authors are referring to in a specific context. Often 'Hg concentrations' are stated when it isn't completely clear which species (GEM, GOM, or PBM) or which measurement technique (Tekran vs passive samplers) is being referred to in that context.

**Response #1**

Thanks for the suggestion. We carefully reviewed the article in relation to "Hg concentrations" and We have carefully polished the language of the manuscript. Given the relatively low accuracy of the data obtained using passive sampling monitoring, they were used only in a very small part of the paper, while the data of concentrations mainly came from Tekran.

**Comment #2**

Throughout the text, the authors write 'under the control of' or 'control period' when referring to transport/circulation patterns. While this is understandable after several readings and sometime thinking about the meaning, this phrasing can be reworded to be more concise and readable. This would go a long way to improving the ease of readability of this manuscript.

**Response #2**

Thanks for the suggestion. We are sorry for the inaccuracies and thank the reviewer for your patience. We have reviewed the description of the atmospheric circulation factors in the article and tried our best to improve the ease of readability of the manuscript. All the revisions have been highlighted in blue in the revised manuscript.

**Comment #3**

The authors make a great effort to characterize the sources and transport patterns of GEM using clustering of back trajectories and PSCF. However, I was quite perplexed to find that no effort had been made to couple back trajectories to GOM or PBM concentrations.

**Response #3**

Thanks for the suggestion. In this manuscript, we carried out trajectory analysis for GEM. Considering the complex topography of the Tibetan Plateau and the fact that most of the trajectories pass through the YZB Grand Canyon, where the subsidence of GOM/PBM is very complex, we think that backward trajectory simulations of GOM and PBM at Nyingchi may introduce considerable errors and uncertainties.

**Comment #4**

The GEM passive samplers data are presented although discussed only briefly. This is an underutilized dataset in this manuscript, the large variations in the data warrant further analysis.

**Response #4**

Thanks for the suggestion. We have extended the discussion of GEM passive sampling data in the revised manuscript. In section 3.1, we added seasonal variation information to the plots of passive sampling data, and added discussions on GEM seasonal variation. The added text is: '**In terms of seasonal variation, average GEM concentrations were the lowest in summer (1.03±0.09 ng m$^{-3}$), with almost identical average concentrations in spring, autumn and winter (1.14±0.28 ng m$^{-3}$, 1.16±0.35 ng m$^{-3}$ and 1.14±0.28 ng m$^{-3}$, respectively). This is in contrast to the trends in the surrounding areas, where the highest GEM concentrations in Nam co, Mt. Ailao, Mt. Waliguan and Mt. Gongga (Yin et al., 2018; Zhang et al., 2016; Fu et al., 2012; Fu et al., 2008) were all found in summer, which may indicate that the Indian summer winds that bring high summer GEM concentrations to these areas do not present similar effect on the SET region.**'
We have also calculated the trajectories for the entire passive sampling period and added discussions of the sources of trajectories for different seasons, as well as discussions of the trajectories for the higher and lower monitored concentrations in the passive sampling period in section 3.3. The added text is: '**We also calculated backward trajectories for the passive sampler monitoring period. Figure S4 shows the trajectories of air masses arriving at the SET station in different seasons. Due to the low accuracy of the data obtained from passive sampling, we didn't combine the GEM concentrations from the passive sampler monitoring with the trajectories here. Except for winter, the vast majority of trajectories originated from the south of the SET station, and most of the trajectories are short in distance. This may be related to the complex local topography, which may also suggest that long-distance transport has limited effect on SET station. There is a partial shift of the backward trajectory from the southwest to the south in spring, compared to summer, which may originate mainly from the influence of the Indian monsoon. The abundance of precipitation, halogens from the Indian monsoon, and rapid growth of vegetation during the monsoon period may have depleted Hg species, and resulted in the lower GEM concentrations in summer. Trajectories from the northern branch of the westerly circulation were more abundant in autumn compared to winter, but did not appear to have an impact on local mean GEM concentrations. Because of the large concentration variations in the passive sampling monitoring, we aggregated the trajectories for the periods of high concentrations (GEM concentrations above 1.5 ng m$^{-3}$) and low concentrations (GEM concentrations below 1.0 ng m$^{-3}$) and performed a cluster analysis. The majority of trajectories in both categories were from the southern part of the SET station and were of similar length (Figure S5), which indicates that the differences in concentrations monitored by passive sampling may not be related to external transport.** '

**Comment #5**
The results of the PCA analysis, at least to me, indicate that long-range transport is the dominant source of GEM while local emissions are more important for GOM and PBM. This is a key result from this study which is listed and mentioned briefly. The author proposes yak dung to be an important local source yet only speculate and do not provide any references that show this could be a source of GOM or PBM. A similar comment for the snowmelt factor, during ISM1, snowmelt is a source of GEM and GOM. From Fig. 2, it appears this factor could be occurring only during a short period (the large spike in GEM and GOM at the end of ISM1), which could be investigated in more detail (e.g., was there snow on the ground during this time, what was the wind direction, temperature, RH, solar radiation during this time?). Expanding on the PCA analysis could give more insight into the local sources of Hg species at Nyingchi.

**Response #5**

Thanks for the suggestion. We have expanded the PCA analysis at the end of section 3.4. The added text is: '**The PCA results provide some new insights into the sources of Hg species. During active monitoring period, long-distance transport of GEM was the main source of SET station and only occurred at PISM and ISM3. Given the low GEM concentrations in ISM1 and ISM2, it is reasonable that PISM and ISM3 are the main long-distance transport periods for GEM. For GOM and PBM, on the other hand, local sources appear to be more important during active monitoring period. This may be related to the fact that GOM and PBM deposit more easily and have complex transport paths to the SET station. The local sources of GOM and PBM are inconclusive. The concentrations of GOM and PBM monitored at the SET station are not high and the local emissions can be assumed to be small. They might come from yak dung burning or other local sources by the local residents (Rhode et al., 2007; Xiao et al., 2015; Chen et al., 2015), or the strong solar radiation and snow surface reaction, which need to be confirmed by further field experimental studies.**'

To the best of our knowledge, there is no data in literature on the species mercury emission of yak dung burning. However, yak dung is a biomass, a metabolic product of yak grazing, and therefore it can be assumed that burning yak dung is similar to burning biomass. Biomass burning is widely recognized a source of atmospheric GOM and PBM (De Simone et al., 2015; De Simone et al., 2016), thus GOM and PBM might also be released during the burning of yak dung.

Regarding the large spike in GOM at the end of ISM1, we have added a discussion at the end of section 3.1. The added text is: '**Table S3 shows the variations of Hg species, meteorological factors and other pollutants from June 1 to 4, 2019. High GOM concentrations were observed on June 2 and 3, and very high solar radiation and UV Index were also observed in these days. PBM concentrations, relative humidity and $O_3$ were low during this period. The solar radiation was nearly twice the mean value of the ISM1 phase (162.79 W m$^{-2}$, Table S2), and thus higher solar radiation might contribute to the higher GOM concentrations. PBM might be partly converted to GOM, but the decrease in PBM concentration was less than the increase in GOM concentration. Generally, high $O_3$ concentrations should be observed at high solar radiation (Kondratyev et al., 1996), but low $O_3$ concentrations were found at Nyingchi, suggesting that $O_3$ may be involved in the formation of GOM. The oxidation of GEM by OH and $O_3$ to generate GOM has been discussed in previous studies with model simulation (Sillman et al., 2007), which may explain the reduced concentration of $O_3$, while OH radicals may be associated with high solar radiation. The mechanism of GOM formation should be further explored in future studies.**'

**Comment #6**

One practical note, please follow ACPs guidelines on the placement of figures and figure captions 'Figures and tables as well as their captions must be inserted in the main text near the location of the first mention (not appended to the end of the manuscript).'. It wasn't practical to change between text and figures, especially when the captions were also in a different location. Also, please put a line between references in the bibliography, it was quite difficult to find a certain reference when they are all bunched together. The references need to be properly formatted as well.

**Response #6**

Thanks for the suggestions. Revisions have been made accordingly.

**Specific Comments**

**Comment #7**

Line 29: I feel there is a better word than 'infected' which can be used here. Possibly 'influenced'.

**Response #7**

We have replaced the word accordingly. Thanks for the suggestion.

**Comment #8**

Lines 33-36: The authors separate the ISM into three periods but list an average for the entire ISM. Maybe it could be beneficial to list averages for all three periods or list the periods in descending order? There is also significant overlap between the standard deviations for parameters between periods. Have the authors performed any statistical tests like a t-test or Wilcoxon Rank Sum test to test for significant differences?

**Response #8**

We have added data on Hg species concentrations for different ISM stages in section 3.1. We didn't add it to the Abstract because it would make the Abstract too long. The GEM and PBM concentrations during the preceding Indian summer monsoon (PISM) period ($1.20\pm0.35$ ng m$^{-3}$, and $11.4\pm4.8$ pg m$^{-3}$ for GEM, and PBM, respectively) were significantly higher than those during the ISM period ($0.95\pm0.21$ ng m$^{-3}$, and $8.8\pm6.0$ pg m$^{-3}$). The GOM concentration during the PISM period ($13.5\pm7.3$ pg m$^{-3}$) was almost at the same level with that during the ISM period ($12.7\pm14.3$ pg m$^{-3}$).

The added text in the Abstract is: '**The GEM and PBM concentrations during the preceding Indian summer monsoon (PISM) period ($1.20\pm0.35$ ng m$^{-3}$, and $11.4\pm4.8$ pg m$^{-3}$ for GEM and PBM, respectively) were significantly higher than those during the ISM period ($0.95\pm0.21$ ng m$^{-3}$, and $8.8\pm6.0$ pg m$^{-3}$). The GOM concentration during the PISM period ($13.5\pm7.3$ pg m$^{-3}$) was almost at the same level with that during the ISM period ($12.7\pm14.3$ pg m$^{-3}$).**'

The added text in section 3.1 is: '**From ISM1 to ISM3, the average GEM concentrations increased from $0.92\pm0.23$ ng m$^{-3}$, $0.92\pm0.18$ ng m$^{-3}$ to $1.04\pm0.21$ ng m$^{-3}$, while GOM concentrations decreased sharply from $18.2\pm29.2$ pg m$^{-3}$, $13.5\pm5.5$ pg m$^{-3}$ to $6.0\pm5.0$ pg m$^{-3}$, and PBM concentrations decreased sharply from $15.4\pm7.9$ pg m$^{-3}$, $7.9\pm3.4$ pg m$^{-3}$ to $3.9\pm3.6$**

**pg m$^{-3}$.'**

**Comment #9**

Lines 36-37: While the passive sampling was for one year, stating the annual average here can be misleading since this information isn't in the abstract. It could also be beneficial to indicate the seasonal averages or variations instead of just an annual average.

**Response #9**

Thanks for the suggestion. We have rewritten this sentence to make it clear. The revised text is:

**'The average GEM concentration in the Nyingchi region was obtained using passive sampler**

**as 1.12±0.28 ng m$^{-3}$ (from April 4, 2019 to March 31, 2020).'**

In section 3.1, we have added seasonal variation to the passive sampling data plots and added a discussion of GEM seasonal variation. The added text is: **'In terms of seasonal variation, average**

**GEM concentrations were the lowest in summer (1.03±0.09 ng m$^{-3}$), with almost identical**

**average concentrations in spring, autumn and winter (1.14±0.28 ng m$^{-3}$, 1.16±0.35 ng m$^{-3}$ and**

**1.14±0.28 ng m$^{-3}$, respectively). This is different from the trends of GEM concentrations in the**

**surrounding areas, where the highest GEM concentrations in Nam co, Mt. Ailao, Mt.**

**Waliguan and Mt. Gongga (Yin et al., 2018; Zhang et al., 2016; Fu et al., 2012; Fu et al., 2008)**

**were all seen in summer, which may indicate that the Indian summer winds that bring high**

**GEM concentrations to these areas do not present similar effect on the SET region.'**

**Comment #10**

Lines 37-38: The authors should indicate the sampling area was clean compared to other high- altitude sites.

**Response #10**

We have added the information in the revised manuscript. Thanks for the suggestion. The revised text is: **'The GEM concentration showed that the sampling area was very clean compared to**

**other high-altitude sites.'**

**Comment #11**

Lines 38-40: These sentences describe only half of the diurnal pattern in the respective periods. It could be beneficial to state other diurnal features present during the different periods. For instance, simply add that during the PISM afternoon concentrations were lower (which is still due to boundary layer dynamics) and that low concentrations of GEM were observed during the morning in the ISM

due to vegetation effects.

**Response #11**

Thanks for the suggestion. We have added the information accordingly. The revised text is: **'Stable**

**high GEM concentrations occur at night and low concentrations occur at afternoon during**

**PISM, which may be related to the nocturnal boundary layer structure. High values occurring**

**in the late afternoon during the ISM may be related to long-range transport. Low**

**concentrations of GEM observed during the morning in the ISM may originate from**
**vegetation effects.'**

**Comment #12**
Line 42: Maybe 'circulation patterns' would fit better here than 'airflow fields'?
**Response #12**
We have replaced the words accordingly. Thanks for the suggestion.

**Comment #13**
Lines 42-43: The authors should indicate that westerly circulation occurs during the PISM.
**Response #13**
Thanks for the suggestion. We have added the information accordingly.

**Comment #14**
Lines 45-47: It would be helpful to know during which periods the different factors were dominant.
**Response #14**
Thanks for the suggestion. We have added the information accordingly. The added text is: '**Long-**
**distance transport factor dominates during PISM and ISM3, while local emissions is the major**
**contributor between PISM and ISM3.'**

**Comment #15**
Line 47: I feel the abstract is missing one sentence stating how this research will be valuable, similar
to the wording on lines 121-122.
**Response #15**
Thanks for the suggestion. We added the following sentence here: '**Our results reveal the Hg**
**species distribution and possible sources of the most important water vapor channel in the**
**Tibetan Plateau, and could serve a basis for further transboundary transport flux**
**calculations.'**

**Comment #16**
Line 50: This sentence requires a reference.
**Response #16**
Thanks for the suggestion. We have added Mason et al., 1994, and Mason et al., 1995 to support
this statement.

**Comment #17**
Line 55: Are GOM and PBM undergoing chemical reactions that lead to their wet and dry deposition?
To my knowledge, this is due to their water solubility (GOM and PBM) and low vapor pressure
(GOM). Maybe the authors could be more specific in their description here.

**Response #17**

Thank you for pointing out the mistake. We have changed the statement in the revised manuscript, as follow: '**In contrast, GOM and PBM are easily removed from the atmosphere through chemical reaction and deposition because of their chemical activity and water solubility, and could therefore bring significant impacts to the local environment (Lindberg and Stratton, 1998; Seigneur et al., 2006)**.'

**Comment #18**

Line 57: 'physicochemical' instead of 'physiochemical'. I also make this mistake which is why I caught it.

**Response #18**

We have replaced the words accordingly. Thanks for the suggestion.

**Comment #19**

Line 60: 'effects'

**Response #19**

We have replaced the words accordingly. Thanks for the suggestion.

**Comment #20**

Line 63-67: I am surprised the Arctic Monitoring Assessment Programme is not listed here (Arctic Monitoring and Assessment Programme | AMAP) as this is an important Hg monitoring network covering North American and European Arctic. Also, it be might be beneficial to the reader if references for individual networks are listed with the acronym, similar to the AMNet.

**Response #20**

Thanks for the suggestions. We have added the Arctic Monitoring Assessment Programme here. References of individual networks are also listed with acronyms in the revised manuscript, as follow: '**The Atmospheric Mercury Network (AMNet; Gay et al., 2013), the Global Mercury Observation System (GMOS; Sprovieri et al., 2013; Sprovieri et al., 2016), the Canadian Atmospheric Mercury Network (CAMNet; Kellerhals et al., 2003) and the Arctic Monitoring Assessment Programme (AMAP; https://mercury.amap.no/) are the main monitoring networks operating in North America and Europe, and the majority of them only monitor GEM concentrations (Gay et al., 2013; Sprovieri et al., 2013; Sprovieri et al., 2016; Kellerhals et al., 2003)**.'

**Comment #21**

Line 66: The semicolon may be removed and replaced with 'and the'. In my opinion, this will improve the readability of the sentence.

**Response #21**

We have replaced it accordingly. Thanks for the suggestion.

**Comment #22**

Lines 80-81: As currently constructed, this sentence isn't representative of the text in Chen et al. (2016). From Chen et al. (2016) 'The total fuel-related atmospheric mercury emissions amount to 859.12 t, to which coal, oil products and biomass contribute 85.77%, 9.06% and 5.17%, respectively.' So, it appears coal contributes 86 % of fuel combustion emissions. This sentence should be reworded to reflect this.

**Response #22**

Thank you for pointing out the mistake. We have changed the statement in the revised manuscript to make it clearer, as follow: '**Considering that coal is the largest emission source of Hg in the atmosphere (approximately 86% of fuel-related atmospheric Hg emissions come from fuel combustion (Chen et al., 2016)), both China and India have great Hg emission potential.**'

**Comment #23**

Line 112: The Tekran speciation units are quite uncertain in terms of collection efficiency (Marusczak et al., 2017; Huang et al., 2017; Gustin et al., 2015), therefore I would recommend removal of the phrase 'high-precision' from this sentence.

Marusczak, N., Sonke, J. E., Fu, X., and Jiskra, M.: Tropospheric GOM at the Pic du Midi Observatory – Correcting Bias in Denuder Based Observations, Environ. Sci. Technol., 51, 863–869, https://doi.org/10.1021/acs.est.6b04999, 2017.

Huang, J., Miller, M. B., Edgerton, E., and Sexauer Gustin, M.: Deciphering potential chemical compounds of gaseous oxidized mercury in Florida, USA, Atmos. Chem. Phys., 17, 1689–1698, https://doi.org/10.5194/acp-17-1689-2017, 2017.

Gustin, M. S., Dunham-Cheatham, S. M., Huang, J., Lindberg, S., and Lyman, S. N.: Development of an Understanding of Reactive Mercury in Ambient Air: A Review, Atmosphere, 12, 73, https://doi.org/10.3390/atmos12010073, 2021.

**Response #23**

Thanks for the suggestion. We agree with the reviewer that 'high-precision' is inappropriate here. We have replaced the phrase 'high-precision' with 'high time resolution'.

**Comment #24**

Line 117: When referring to 'cluster analysis', do the authors mean PCA or clustering of back trajectories?

**Response #24**

Thanks for the comment. It's the cluster analysis of back trajectories. We have changed the statement in the revised manuscript to make it clearer, as follow: '**To better identify the sources of Hg pollution and potential pollution areas, we combined real-time GEM monitoring data with backward trajectory analysis, and a follow-up cluster analysis of back trajectories.**'

**Comment #25**

Line 119: 'sources'

**Response #25**

We have replaced it accordingly. Thanks for the suggestion.

**Comment #26**

Line 131: It could be helpful to the reader if the authors state the temperature for the PISM and the ISM since the manuscript revolves around these periods.

**Response #26**

Thanks for the suggestion. We have added the information accordingly. '**The average annual air temperature is 5.6 °C, the average air temperature during PISM and ISM periods are 6.0 °C and 12.0 °C, respectively.**'

**Comment #27**

Line 134: Other than the YZB Grand Canyon, what are the other water vapor channels?

**Response #27**

Many studies of the water vapor pathway have concluded that YZB Grand Canyon is the only major water vapor transport channel on the southern Tibetan Plateau (Ping and Bo, 2018; Yan et al., 2020; Gong et al., 2019b; Feng and Zhou, 2012).

**Comment #28**

Lines 134-135: Similar comment as above but for precipitation.

**Response #28**

Many studies of the water vapor pathway have concluded that YZB Grand Canyon is the only major water vapor transport channel on the southern Tibetan Plateau (Ping and Bo, 2018; Yan et al., 2020; Gong et al., 2019b; Feng and Zhou, 2012).

**Comment #29**

Line 141: Can the authors give some examples of this unique high-altitude distribution pattern of biomes and vegetation in the area? This would aid the reader and help explain the interpretation that vegetation effects have a significant effect on GEM concentrations.

**Response #29**

Thanks for the suggestion. We have added some information accordingly. '**Interactions between terrestrial ecosystems and atmosphere have contributed to the development of diverse biomes and distinctive vegetation elevation distribution patterns from tropical rainforests to boreal forests and tundra.**'

**Comment #30**

Line 149: These dates are different from the ones listed in the abstract.

**Response #30**

Thanks for pointing out the mistake. We have re-examined the data and made revisions. The correct deployment time should be from March 30 to September 3, 2019, as described in the abstract.

**Comment #31**

Line 155: 'drawn in' instead of 'sucked' and 'into' instead of 'in'.

**Response #31**

We have replaced it accordingly. Thanks for the suggestion.

**Comment #32**

Lines 157-160: Having worked with the Tekran instruments, I understand what is meant when the authors describe the sample collection procedure, however, a reader unfamiliar with this procedure could misinterpret the text. The time required to collect and analyze one sample is two hours, one hour for collection and one hour for analysis. This isn't stated clearly here, I suggest rephrasing these sentences to make this clearer to the reader.

**Response #32**

Thanks for the suggestion. We have changed the description of the sample collection procedure in the revised manuscript to make it clearer. The revised text is: '**A complete measurement cycle**

**takes two hours. During the first hour, GOM was enriched on a KCL-coated annular denuder,**

**PBM was enriched on a quartz fiber filter (QFF), and GEM was directly enriched on the gold**

**tube of the Tekran 2537B and measured directly by cold vapor atomic fluorescence**

**spectroscopy (CVAFS). The collected PBM and GOM were desorbed in succession to Hg(0)**

**at temperatures of 800 ℃ and 500 ℃ in the following hour, respectively. Then the Hg(0) was**

**measured by Tekran 2537B.**'

**Comment #33**

Lines 165-167: Can the authors elaborate on the method from Slemr et al. (2016)?

**Response #33**

Thanks for the suggestion. According to Slemr et al. (2016), the small captured Hg amount would probably cause the bias of the measurement. Considering the high altitude at which the instrument was installed, as well as to mitigate the impacts of low atmospheric pressures on the pump's operation, a low air sampling rate of 7 L min$^{-1}$ for the pump model and 0.75 L min$^{-1}$ (at standard pressure and temperature) for model 2537B were applied in this study. We have used the function given in Figure 3 in Slemr et al. (2016) to correct the data obtained from the monitoring.

**Comment #34**

Line 170: Again, these dates are different from the abstract. These dates need to be reconciled. Also, why is a day not stated here when it is other places.

**Response #34**

Thanks for pointing it out. The sampling period of passive samplers was from April 4, 2019 to
March 31, 2020. We have added the date to the abstract.

**Comment #35**

Lines 173-174: The authors need to state a more precise sampling interval for the passive samplers.

**Response #35**

Thanks for the suggestion. The sampling intervals for the passive samplers were close to once a month from April 4 to July 10, 2019, and three times a month from July 10, 2019 to March 31, 2020.

We have added detailed start and finish times for every sampling period in the support information.

**Comment #36**

Line 175: What is a DMA-80? Can the authors give more information on this instrument?

**Response #36**

Thanks for the suggestion. We have added more information about DMA-80 in the revised manuscript. We also provided our previous studies as a reference with detailed information on laboratory analysis procedures. '**DMA-80 is an instrument that was used in accordance with US**

**EPA Method 7473, using a combined sequence of thermal decomposition, mercury**

**amalgamation and atomic absorption spectrophotometry (Zhang et al., 2012).**'

**Comment #37**

Line 199: Would 'air parcels' be a better term than 'matter' in this context?

**Response #37**

We have replaced it accordingly. Thanks for the suggestion.

**Comment #38**

Lines 202-203: What is the typical boundary layer height at Nyingchi? Are there times when the boundary layer is below 1000 m? Have the authors varied the arrival height to see its effect on air mass origin? Have the authors calculated trajectories longer than 72 hours? For GOM and PBM, this length is reasonable, however, for GEM the lifetime is much longer and could be affected by sources further away than 72 hours. While the input meteorological data is at a time resolution of 6

h, the HYSPLIT model can interpolate these data and produce hourly trajectories. This would increase the uncertainty but would allow for measurements of GOM and PBM to be integrated with these trajectories. Have the authors explored such an analysis? Do the authors mean 'simulated'

instead of 'stimulated'?

**Response #38**

Thanks for the suggestion. The relatively high trajectory arrival height was set mainly due to concerns that the complex topography of the Tibetan Plateau might cause significant disruptions to the trajectory. We reviewed the data and found out that the average boundary layer height in

Nyingchi is 457 m (data from Global Data Assimilation System (GDAS)). In the revised manuscript, we have recalculated all trajectories and redo all the simulations associated with the trajectories.
The arrival height was set at 200 m a.g.l., which is about half of the boundary layer height.
Considering that a longer simulation time will bring higher trajectories uncertainty, and 120 hours
are sufficient for trajectories transmission over longer distances, every backward trajectory was
simulated for 120 hours at 3 hours intervals. Also, we examined the effect of arrival height on the
trajectories using different arrival heights (20m, 50m, 200m and 500m, respectively) in June 2019.
The results show that the calculated trajectories of the air masses are almost the same when the
arrival height is below 500m. The figure below shows the trajectories to Nyingchi in June 2019 with
different air masses arrival heights. We also added the results in the support information in the
revised manuscript.

[Figure]

Figure Trajectories to Nyingchi in June 2019 with different air masses arrival heights

In this manuscript, we only carried out trajectory analysis for GEM. Considering the complex
topography of the Tibetan Plateau and the fact that most of the trajectories pass through the YZB
Grand Canyon, where the subsidence of GOM or PBM is more complex, we think that backward
trajectory simulations of GOM and PBM at Nyingchi may introduce considerable errors. We hope
that future work could help identify the transport behavior and speciation transformations of GOM
and PBM through more refined simulations and more observational data.
We have replaced 'stimulated 'with 'simulated' accordingly. Thanks for pointing out the mistake.
We have changed the description of the backward trajectory simulations in the revised manuscript
to make it clearer. The revised text is: '**The trajectory arrival height was set to 200 m a.g.l., which**
**is about half of the boundary layer height. We examined the effects of arrival height on the**
**trajectories using different arrival heights (20m, 50m, 200m and 500m respectively) in June**
**2019. The results show that the calculated trajectories of the air masses are almost the same**

[revised manuscript text omitted]

**Comment #39**

Lines 204-205: The last sentence in this paragraph needs to be reworded.

**Response #39**

Thanks for the suggestion. We have reworded the sentence as follow '**Cluster analysis can help identify the average air masses transport path by averaging similar or identical paths in the existing air masses paths, and provide major directions of GEM transported to the measurement site.'**

**Comment #40**

Lines 206-212: The description of PSCF needs to be expanded. What was the threshold percentile? What was the arbitrary weighting function used? These parameters need to be stated for this research to be reproducible.

**Response #40**

Thanks for the suggestion. We agree with the reviewer that PSCF analysis couldn't provide gainful information in this manuscript. We have decided to delete the PSCF related discussion.

**Comment #41**

Lines 218-222: Can the authors elaborate on the tests and procedures used for determining the optimal solution for the PCA analysis? For example, what are the Kaiser-Meyer-Olkin measure of sampling adequacy and Bartlett's test of sphericity used for? What was the outcome? Please define MSA. Were there multiple elbows in the scree plots?

**Response #41**

Thanks for the suggestion. The Kaiser-Meyer-Olkin measure of sampling adequacy (>0.5) and Bartlett's Test of sphericity ($p < 0.05$) tests are used to determine that PCA is a suitable method for the data set. This test is to ensure that the PCA has been used correctly and to guarantee the reliability of the analysis results. MSA is an abbreviation of measure of sampling adequacy. In our analysis process, there is only one obvious elbow in every scree plot. We have revised the manuscript to make it clear, as follow: '**To ensure that the PCA is a suitable method for the data set in this study, the Kaiser-Meyer-Olkin measure of sampling adequacy (> 0.5) and Bartlett's test of sphericity ($p < 0.05$) tests were performed in the initial PCA run.**'

**Comment #42**

Line 228: The text states 'daily' here and in other places, but the rightmost y-axis label in Fig. 2 gives units of 'nm 2 hour'. Can the authors please clarify this discrepancy?

**Response #42**

Thanks for pointing out the mistake. We reviewed the rainfall data and found that the rainfall resolutions are 2 hours. We have deleted 'daily' in the revised manuscript accordingly. The title of Figure 2 has also been revised.

**Comment #43**

Lines 231-232: What are the criteria for dividing the ISM into three periods in terms of precipitation? Please elaborate on these criteria and the reasoning behind the selection of the timing of the different periods.

**Response #43**

Thanks for the suggestion. The ISM period was further subdivided into three periods (ISM1 – ISM3). However, there is no strict criteria for the selection of the timing of the different periods. We made a rough division based on the changes of precipitation and the development of the monsoon.

**Comment #44**

Lines 232-235: Please see my comments about listing the concentrations for different ISM periods from the abstract.

**Response #44**

Thanks for the suggestion. We have listed average concentrations of GEM, GOM, PBM for all three periods in the revised manuscript. We also provided statistics metrics of Hg species, meteorological factors and other pollutants for all periods in the support information, as follows: '**From ISM1 to ISM3, the average GEM concentrations increased from 0.92±0.23 ng m$^{-3}$, 0.92±0.18 ng m$^{-3}$ to 1.04±0.21 ng m$^{-3}$, while GOM concentrations decreased sharply from 18.2±29.2 pg m$^{-3}$, 13.5±5.5 pg m$^{-3}$ to 6.0±5.0 pg m$^{-3}$, PBM concentrations decreased sharply from 15.4±7.9 pg m$^{-3}$, 7.9±3.4 pg m$^{-3}$ to 3.9±3.6 pg m$^{-3}$.**'

**Comment #45**

Line 235: I think the words 'locally monitored' can be omitted.

**Response #45**

Thanks for the suggestion. We have deleted it accordingly.

**Comment #46**

Line 237: Same but for 'decisive'.

**Response #46**

Thanks for the suggestion. We have deleted it accordingly.

**Comment #47**

Line 243: I feel there is a better reference for the chemical properties of GEM than Horowitz et al. (2017), which deals with modeled redox chemistry of Hg. Possibly a review paper, or references from a review paper, might be more appropriate here.

**Response #47**

Thanks for the suggestion. We have changed the reference (Selin, 2009).

**Comment #48**

Lines 244-246: Is this total precipitation or an average during these periods? It is interesting that GOM decreased by roughly half while PBM only decreased by ~25 %.

**Response #48**

Thanks for pointing out the mistake. It is total precipitation in the monitoring station during these periods, and we have revised it to make it clear. We also found that the concentrations of GOM and PBM have been listed in the wrong order. Actually, the GOM decreased by ~25 % while PBM decreased by roughly half. Revisions are as follow: '**With the increase in rainfall from 113.75 mm during ISM1 period to 373.28 mm during ISM2 period (total precipitation), the concentrations of GOM and PBM decreased sharply from 18.2±29.2 pg m$^{-3}$ and 15.4±7.9 pg m$^{-3}$ to 13.5±5.5 pg m$^{-3}$ and 7.9±3.4 pg m$^{-3}$, respectively.**'

**Comment #49**

Lines 249-252: This is an important result of a previous study. During the PISM, GEM is mainly from long-range transport, while during the ISM local emissions is an important source of GOM and PBM (from the PCA analysis). These local emissions could be important for total Hg in rainwater.

We agree with the reviewer that the local emissions could be important for total Hg in rainwater during ISM period. We have added a discussion about local emissions in the revised manuscript.

**Comment #50**

Line 255-258: It was stated in the site description that westerly circulation patterns are dominant from September to April and that ISM circulation patterns are dominant from May to August. Was this information obtained through trajectory analysis or previous knowledge from the site? This information is again presented here and used to explain the higher passive sampler GEM concentrations in the later part of the sampling period. I am curious if any trajectories were calculated for the passive sampler period? This could be used to directly support the abovementioned statements. The large variations in the passive sampler period, in my opinion, warrant further investigation. What were the meteorological conditions or transport patterns under high and low concentrations?

**Response #50**

Thanks for the comments and suggestions. The Asian summer monsoon and the mid-latitude Westerlies are major atmospheric circulation systems influencing the climate of the Tibetan Plateau, which could be seen in previous studies (Yao et al., 2013; Benn and Owen, 1998; Kotlia et al., 2015; Sun et al., 2020; Liu et al., 2016; Huang et al., 2013). The Indian Monsoon Index can be used to determine the onset of the summer monsoon. We have added the Indian Monsoon Index for 2019 in the supporting information (Figure S1), with the Indian monsoon starting to break out in May, 2019 and becoming the dominant wind field. We also calculated the trajectories for the entire passive sampler period, and added a discussion of the sources of trajectories for the different seasons and a discussion of the trajectories for the higher and lower monitored concentrations in the passive sampler period in section 3.3. The added text is: '**We also calculated backward trajectories for the passive sampler monitoring period. Figure S4 shows the trajectories of air masses arriving at the SET station in different seasons. Due to the low accuracy of the data obtained from passive sampling, we didn't combine the GEM concentrations from the passive sampler monitoring with the trajectories here. Except for winter, the vast majority of trajectories originated from the south of the SET station, and most of the trajectories are short in distance. This may be related to the complex local topography, which may also suggest that long-distance transport has limited effect on SET station. There is a partial shift of the backward trajectory from the southwest to the south in spring, compared to summer, which may originate mainly from the influence of the Indian monsoon. The abundance of precipitation, halogens from the Indian monsoon, and rapid growth of vegetation during the monsoon period may have depleted Hg species, and resulted in the lower GEM concentrations in summer. Trajectories from the northern branch of the westerly circulation were more abundant in autumn compared to winter, but did not appear to have an impact on local mean GEM**

**concentrations. Because of the large concentration variations in the passive sampling monitoring, we aggregated the trajectories for the periods of high concentrations (GEM concentrations above 1.5 ng m$^{-3}$) and low concentrations (GEM concentrations below 1.0 ng m$^{-3}$) and performed a cluster analysis. The majority of trajectories in both categories were from the southern part of the SET station and were of similar length (Figure S5), which indicates that the differences in concentrations monitored by passive sampling may not be related to external transport.** '

**Comment #51**

Lines 258-260: I agree this is most likely the case, given the Hg emission inventory and trajectory clusters plotted in Fig. 5. Calculating trajectories for the entire passive sampler period would directly show this.

**Response #51**

We have calculated trajectories for the entire passive sampler period and added a discussion of the sources of trajectories for the different seasons and a discussion of the trajectories for the higher and lower monitored concentrations in the passive sampler period in section 3.3.

**Comment #52**

Lines 260-262: This is nice since it gives the reader context, however, maybe it would benefit the reader to move it to the beginning of this paragraph.

**Response #52**

Thanks for the suggestion. We agree with the reviewer that it should be more appropriately placed at the beginning of the paragraph.

**Comment #53**

Line 272: Is there a better way to say 'monsoon control zones'? See general comments above.

**Response #53**

Thanks for the suggestion. We have revised the presentation and carefully revised other relevant presentations throughout the text.

**Comment #54**

Line 276: I feel there is a better phrase than 'violent' to describe depositional processes. Possibly 'extreme'?

**Response #54**

Thanks for the suggestion. We agree that 'extreme' is better here.

**Comment #55**

Lines 283-284: 'generally believed' isn't the most appropriate language for a scientific article. Please rephrase.

**Response #55**

Thanks for the suggestion. We have revised as follow: 'Previous studies (Lin et al., 2019; Gong et al., 2019a; Wang et al., 2015) indicated that pollutants from the heavily polluted Indian subcontinent may be transported to the Tibetan Plateau under the action of ISM, resulting in increased local pollutant concentrations on the plateau.'

**Comment #56**

Line 290: Fu et al. (2016) provide an excellent explanation of the decrease of GEM over the whole ISM and the diurnal profile at night. However, this study was conducted in a different geographical region and at a lower altitude. Can the authors offer any reasoning for why this effect is valid at both locations? For instance, is there similar vegetation at both sites?

**Response #56**

The forest in Fu et al. (2016) is dominated by *Pinus koraiensis, Fraxinus mandshurica, Tilia amurensis, Acer mono and Quercus mongolica*. In the YZB Grand Canyon, interactions between terrestrial ecosystems and the atmosphere have contributed to the development of diverse biomes and distinctive vegetation elevation distribution patterns from tropical rainforests to boreal forests and tundra. Major tree species in Fu et al. (2016) can be found in the YZB Grand Canyon. So we believed that the effect is also valid at the Grand Canyon.

**Comment #57**

Line 291: This is also a very logical explanation for the decrease in GEM during the ISM, however, this statement requires a reference. Have other locations in India observed enhancements of halogens during the ISM?

**Response #57**

Thanks for the suggestion. We have added a reference (Fiehn et al., 2017) here.

**Comment #58**

Lines 291-293: From Fig. 2, it appears that during the beginning of ISM1 GOM concentrations are lower than ISM2 and on a similar level to ISM3. However, there is alarge spike in GOM at the end of ISM1 that could be skewing the average for this period. Has this spike in GOM been investigated in more detail?

**Response #58**

Thanks for the comment. It is an interesting phenomenon. We have added a discussion at the end of section 3.1, as follow: '**Table S3 shows the variations of Hg species, meteorological factors and other pollutants from June 1 to 4, 2019. High GOM concentrations were observed on June 2 and 3, and very high solar radiation and UV Index were also observed in these days. PBM concentrations, relative humidity and O$_3$ were low during this period. The solar radiation was nearly twice the mean value of the ISM1 phase (162.79 W m$^{-2}$, Table S2), and thus higher solar radiation might contribute to the higher GOM concentrations. Some of the PBM might be**

converted to GOM, but the decrease in PBM concentration was less than the increase in GOM concentration. Generally higher $O_3$ concentrations should be observed at higher solar radiation (Kondratyev et al., 1996), but lower $O_3$ concentrations were found at Nyingchi, suggesting that $O_3$ may contribute to the formation of GOM. The oxidation of GEM by OH and $O_3$ to generate GOM has been discussed in previous studies in model simulations (Sillman et al., 2007), which may explain the reduced concentration of $O_3$, while OH radicals may be associated with higher solar radiation. The mechanism of GOM formation should be further explored in future studies.'

**Comment #59**

Line 297: 'deposit' instead of 'settle' since you are referring to wet deposition.

**Response #59**

Thanks for the suggestion. We agree that 'deposit' is better here.

**Comment #60**

Figure 4: It is impossible to extract information from these figures. Seven axes on one figure are way too many. The lettering for each panel is also very large compared to the figures themselves. The combination of lines with errors represented by dashed lines and dots of small sizes and similar colors is dizzying and makes interpretation unnecessarily difficult. I do not understand why so many parameters are presented when only the Hg species are discussed briefly in the text.

I would suggest either group the Hg species and meteorological parameters separately or group parameters with a similar diurnal profile together. I would then opt for the former and put the diurnal profile of meteorological parameters in the supplement.

**Response #60**

Thanks for the suggestion. We agree with the reviewer that the figures contain too much information. We have redrawn the diurnal variation figures by keeping only GEM and error range, GOM, PBM and wind speed information in the figure.

**Comment #61**

Line 314: Any statement that mentions 'previous research' requires references and citations, both of which are missing from this sentence.

**Response #61**

Thanks for the suggestion. We have added some references accordingly. We also checked for similar problems throughout the article.

**Comment #62**

Lines 323-325: Can the authors expound upon this speculation? They have offered yak dung as a possible source of local emissions elsewhere in the text, is there any other possible local sources of Hg that could explain this observation?

**Response #62**

Thanks for the suggestion. There is no evidence that yak dung is the major reason of the higher GOM concentrations during ISM1. Firstly, from PISM to ISM1, the total amount of yak dung used by residents is decreasing due to the increase in air temperature; Secondly, the Nyingchi area is sparsely populated and the emissions from yak dung should be small. More field studies in the future are needed to provide more accurate explanation.

As the discussion we added in the last paragraph of section 3.1, we suggested that higher concentrations of GOM are more likely to be related to the widespread local glacier, higher solar radiation and $O_3$ concentrations, but there is currently insufficient evidence to support this claim. We have added a short discussion here, as follows: '**The oxidation of GEM by OH and $O_3$ to generate GOM may be a possible reason for the high GOM concentration (Sillman et al., 2007). However, the mechanism of GOM formation should be further explored.**'

**Comment #63**

Lines 330-331: I am not sure what is meant by 'chemical dissipation', and there was nothing in the references given. Do the authors mean chemical reactions? Also, the references don't support the statements in the sentence.

**Response #63**

Thanks for pointing out the mistake. We have rewritten this sentence as follow: '**The decrease in GEM concentration at night may be due to the interaction of pollutants from regional emissions and long-range transport (Fu et al., 2008; Fu et al., 2010).**'

**Comment #64**

Line 346: Holmes et al. (2010) isn't an appropriate reference for the reduction of GOM in local snowy mountains. Is there not more specific studies (possible lab or field campaigns) that show this mechanism in more detail?

**Response #64**

Thanks for the suggestion. We have replaced the reference with '(Lalonde et al., 2003; Lalonde et al., 2002)'.

**Comment #65**

Lines 346-347: What do the authors mean by 'field GEM source'?

**Response #65**

Thanks for the comment. We have rewritten it as follow: '**The gradual increase in GEM concentration during the daytime may be due to the reduction of GOM from nearby local snowy mountains (Lalonde et al., 2003; Lalonde et al., 2002) or long-range transported GEM brought in by airflow (Lin et al., 2019).**'

**Comment #66**

Lines 349-350: Please provide references for the Indian Ocean being a source of halogens.

**Response #66**

Thanks for the suggestion. We have added '(Fiehn et al., 2017)' here as a reference.

**Comment #67**

Figure 5: Making the size of the cluster trajectory is a very nice way of intuitively showing the relative proportion of each cluster occurrence, however, it is difficult to grasp the absolute percentage from the legend (this is just an observation not necessarily a suggestion to change it).

Starting the cluster index at zero is a matter of taste, but it is intuitively easier to understand when the index starts at one.

A color scale or color bar is required for the emissions inventories.

Having all the color scales for GEM the same might make it easier to notice the differences between different periods

**Response #67**

Thanks for the suggestion. We have redrawn the trajectory and retained the trajectory size settings.

We have also detailed the cluster number, GEM concentration and ratio on the trajectory edges. We started the cluster index at one in the revised manuscript. A color scale has been added for the emission inventories. The trajectories color setting has been removed in the new version.

**Comment #68**

Line 360: This sentence needs to be reworded. See general comments above.

**Response #68**

Thanks for the suggestion. We have reworded it to make it clear, as follow: '**During this period,**

**the meteorological factors at Nyingchi were mainly controlled by westerly circulation.**'

**Comment #69**

Line 365: 'relatively'.

**Response #69**

Thanks for the suggestion. We have replaced the word accordingly.

**Comment #70**

Line 367-369: This information about the cluster turning in the Bay of Bengal is not represented in the cluster average. It might be beneficial to show the individual trajectories for each cluster in the supplement. Also, as currently constructed, the citation to the UNEP reports appears to reference the turn in trajectories. I suggest moving the citations to the end of the sentence, this would alleviate any confusion.

**Response #70**

Thanks for the suggestion. We have deleted the discussion about the Bay of Bengal accordingly.

Showing the individual trajectories for each cluster will not display valid information because there are too many trajectories. The reference has been moved to the end of the sentence accordingly.

**Comment #71**

Lines 370-372: This is true for GOM and PBM, however, not for GEM, which as stated above in the text, isn't very water-soluble. This is an example, where specifying which Hg species the authors are referring to would lessen any confusion from the reader's perspective.

**Response #71**

Thanks for the suggestion. We have deleted it accordingly. We carefully reviewed the article in relation to "Hg concentrations" and we have carefully polished the language of the manuscript. The trajectory simulation is performed for GEM only, as we have hinted at the beginning of the section:

'**To further investigate the contributions of different sources to the SET site, air mass back**

**trajectory simulation and trajectory cluster analyses were performed for GEM.**'

**Comment #72**

Lines 374-375: Showing the individual trajectories for each cluster during this period would directly show what the text is stating, as right now, the statement is not evident from Fig. 5b.

**Response #72**

Thanks for the suggestion. Showing the individual trajectories for each cluster will not display valid information because there are too many trajectories. We have reworded this sentence as follow:

'**The clusters undergo a slight counter-clockwise rotation.**'

**Comment #73**

Lines 377-378: HYSPLIT can output precipitation and H2O mixing ratio at each trajectory step, this information would show what the authors are suggesting — water vapor is increased when air masses arrive from the Indian Ocean.

**Response #73**

Thanks for the suggestion. We have changed it as: '**With the development of the Indian monsoon,**

**it brings an abundance of water vapor (Ping and Bo, 2018).**'

**Comment #74**

Lines 383-386: A color bar for the Hg emission inventories would be helpful here.

**Response #74**

Thanks for the suggestion. A color bar has been added accordingly.

**Comment #75**

Line 391: De Simone et al. (2015) is about modeled Hg emissions from biomass burning and not with anthropogenic emissions. The UNEP reports seem like a better reference for this statement.

**Response #75**

Thanks for the suggestion. We have changed the citation accordingly.

**Comment #76**

Line 393: It would be more appropriate to list the references given in Lin et al. (2019) for yak dung burning instead of just Lin et al. (2019). I wonder why these references were not given in other locations where yak dung is mentioned. The words 'yak dung' does not appear in Huang et al. (2016). Also, the reference for Lin et al. (2019), Lines 730- 733, appears to be incorrectly formatted.

**Response #76**

Thanks for the suggestion. We have updated the references for yak dung burning here and elsewhere.

**Comment #77**

Line 402: Which species of Hg?

**Response #77**

The trajectory simulation is performed for GEM only. We have deleted this sentence in the revised version.

**Comment #78**

Line 407: Can the authors show that many wildfires existed during this period?

**Response #78**

Thanks for the comment. Since we have recalculated the trajectory, the geographical area covered by the trajectory has been changed.

**Comment #79**

Line 410: This is an example of how the phrasing 'controlling the region' needs to be rewritten to describe the transport patterns and air mass circulation.

**Response #79**

Thanks for the suggestion. We have revised the presentation and carefully revised other relevant presentations throughout the text.

**Comment #80**

Line 412: The cluster average does not show this and traj0 is hardly visible. Interestingly, traj1 appears to have the highest concentrations of GEM and arrives from areas with high Hg emissions but is not mentioned in the text. This cluster occurs rather infrequently though. I agree the weakening of the ISM is likely the reason for the increasing pattern in GEM during the ISM3, but this should at least be mentioned.

**Response #80**

Thanks for the suggestion. We have reselected the trajectory size in the revised manuscript to avoid occlusion. There is no cluster like traj1 in the new clusters.

**Comment #81**

Line 419: Again, I wouldn't refer to measurements made with the Tekran systems as 'detailed'. The exact chemical identify of GOM and PBM is unknown. Therefore, I would remove this word.

**Response #81**

We have removed the words accordingly. Thanks for the suggestion.

**Comment #82**

Lines 418-427: In the previous paragraphs in this section, the authors examine the source regions of GEM and transport patterns during different periods. This PSCF muddles this analysis and do not provide any additional or useful information. The PSCF was applied to GEM, please indicate which species of Hg is being referred to here. The smoothing applied to these figures could be obscuring the analysis. The authors discuss depositional processes during transport affecting Hg concentrations, although this would apply to GOM and PBM and not so much GEM. In my opinion, I would omit the PSCF analysis, as it does not provide gainful information, is not described adequately in the methods section, and contradicts the previous analysis of GEM with trajectory cluster analysis. This is, however, only my opinion.

**Response #82**

Thanks for the suggestion. We agree with the reviewer that the PSCF analysis does not provide gainful information in this manuscript. We have decided to delete the PSCF related discussion.

**Comment #83**

Lines 429-430: I am confused by the number of factors for each period. For example, from Table 2 there are only two factors that occur during the PISM (long-distance transport and local emissions). There is only one factor that is unique to a period (melt during ISM1) and only local emissions occur during all periods. Please clarify this in the text.

**Response #83**

Thanks for the comment and suggestion. As we mentioned at the beginning of section 3.4, 4-5 factors were found for each period from PISM to ISM3 periods, so there were 19 factors in total. For example, in the analysis for ISM1, 5 factors were found and four of them were considered as important Hg-related components because of higher factor loadings. Two of them were assigned to local emissions. We further clarify it as follow: '**Only Hg-related components were reserved here and four underlying PCA factors are summarized (Table 2).**'

**Comment #84**

Table 2: The caption for Table 2 needs to be expanded. I can see that numbers in bold indicate a loading over 0.5, this needs to be stated in the caption. Why are certain species omitted from the PCA analysis for certain periods? This was not clear from the methods section. Why is there two ISM1 for local emissions? Please define VE. Would it be possible to remove the underscores from the column headers?

**Response #84**

Thanks for the suggestion. Table 2 lists the four underlying PCA factors for important Hg-related components. For readability, variables with very low factor loadings (<0.1) are not shown in the Table. As we mentioned at the beginning of section 3.4, 4-5 factors were found for each period from PISM to ISM3, and there were 19 factors in total. In the analysis for ISM1, five factors were resolved and four of them were considered as important Hg-related components because of high factor loadings. Two of them were assigned to local emissions. The classification is proposed mainly based on the distribution characteristics of the factor loadings for other meteorological conditions and pollutant species. VE is an abbreviation of Variance Explained, we have changed it to full spelling in the revised manuscript. In the revised manuscript, we have added a note under Table 2. '**Note: Variables with high factor loadings (> 0.5) were marked in bold. For readability, variables with very low factor loadings (<0.1) are not presented.**'

The underscores from the column headers have been removed accordingly.

**Comment #85**

Line 452: A reference is required for this statement.

**Response #85**

Thanks for the suggestion. We have added (Rhode et al., 2007; Xiao et al., 2015; Chen et al., 2015) in the revised manuscript.

**Comment #86**

Lines 453-462: While meteorology is no doubt affecting the behavior of atmospheric mercury, I am confused about how this factor affects mercury at Nyingchi. A different Hg species are excluded from the PCA for ISM1-3 and the only significant variable is GEM during ISM2. It is not clear from the text how meteorology is affecting GEM during this period.

**Response #86**

Thanks for the comment. These factors have been assigned as meteorological factors because of similar meteorological factor loading distributions. Different Hg species are excluded from the PCA for ISM1-3 because of the lower factor loading rather than artificial selection.

**Comment #87**

Lines 464-467: Please indicate which period the authors are referring to here as well as the panel in Fig. 3. These two sentences largely say the same thing and cite the same studies, one could reasonably combine them for brevity.

**Response #87**

Thanks for the suggestion. We have revised these two sentences, as follow: '**The influence of increasing solar radiation may reflect the snow/ice melt process. which have been proved to be able to increase atmospheric GEM concentration (Huang et al., 2010; Dommergue et al., 2003).**'

**Comment #88**

Lines 469-470: Which 'previous simulations'? Please provide a reference. Are the authors referring to Song et al. (2018)? If so, please cite them or combine this sentence which the previous one. Also, the wording 'previous simulations….during the ISM1 period' implies that simulations were performed for GOM during this campaign. Please rectify this.

**Response #88**

Thanks for the comment and suggestion. We have reorganized the sentences as follows: '**GEM may originate from the evaporation of snow melting and/or be driven by the photoreduction of snow Hg$^{II}$ (Song et al., 2018). The simulation indicated that the oxidation of GEM may occur at the snow/ice interface in the action of solar radiation, and may lead to extra GOM release.**'

**Comment #89**

Line 477: Please see my previous comment about the phrasing 'generally believed'.

**Response #89**

Thanks for the suggestion. We have reworded it accordingly.

**Comment #90**

Line 480: 'masses' instead of 'mass'.

**Response #90**

We have replaced the word accordingly. Thanks for the suggestion.

**Comment #91**

Line 497: Can the authors provide direction or recommendations for further studies?

**Response #91**

Thanks for the suggestion. We believe that additional wet deposition monitoring along the YZB Grand Canyon in the future may provide more evidences on the transportation mechanisms. We have revised the sentence, as follow: '**The deposited pollutants may flow into the downstream area via rivers to Southeast Asia and South Asia. Additional wet deposition monitoring along the YZB Grand Canyon in the future may provide more evidences on transportation mechanisms.**'

**Comment #92**

Line 502: Similar comment as the previous one.

**Response #92**

Thanks for the suggestion. We have revised the sentence, as follow: '**The high GEM concentration during the PISM period may indicate that a large amount of external Hg entered the Nyingchi area during the non-ISM period, and thus monitoring of isotopic atmospheric Hg in future studies or accurate model simulations are needed to provide better evidences.**'

**Comment #93**

Lines 503-511: In combination with the previous study from Qomolangma, this study provides important insights into the transport, dynamics, and processes affecting Hg species during the PISM and ISM. I feel that since these two studies are the first in this geographical area, there should be more of a discussion between the differences and similarities between these two sites. The authors mention differences but only briefly.

**Response #93**

Thanks for the suggestion. We have rewritten and expanded the discussion, as follow: '**The results of our previous study on Qomolangma were different from those in Nyingchi. Qomolangma site locates on the northern side of the Himalayas, a typical terrain on the southern edge of the Tibetan Plateau. The Nyingchi site locates in a typical pathway for air masses to enter the Tibetan Plateau. Both sites locate in sparsely populated areas, far from human activity, making them ideal clean locations to study the behavior of Hg species. Hg species monitoring in both sides could help explain the possible transboundary transport patterns. In terms of the concentration distributions of Hg species, both sites showed low concentrations, with slightly higher GEM concentrations identified at Qomolangma site. The diurnal variations in the concentrations of Hg species are unique in both areas, as there are relatively little anthropogenic disturbances, but Nyingchi is surrounded by greater elevation variation and more complex terrain, and thus the diurnal variation is subject to more natural disturbance factors. In terms of Hg species from long-range transport, Qomolangma was mainly affected by monsoonal transport from India during the ISM period, showing the increases in the concentrations of GEM. Nyingchi, on the contrary, has low GEM concentrations during the ISM. Although receiving almost the same monsoonal influences from India, the intensity of the transport and the subsidence on the transport path may be responsible for the large differences in the concentrations of Hg species and their environmental behavior between the two sites. Together, they represent two typical transboundary transport patterns of Hg in the Tibetan Plateau.** '

**Comment #94**

Conclusions: The Conclusions section is very similar to the Abstract. Please see my Specific Comments from the Abstract section for suggestions and General Comments for topics that should be **highlighted** or discussed in **greater detail**, which should be represented in a revised Conclusions sections.

**Response #94**

Thanks for the suggestion. We have rewritten the Conclusions section, as follow: '**Comprehensive Hg species monitoring was carried out in Nyingchi, a high-altitude site in the southeast of the Tibetan Plateau. Nyingchi is located on the main pathway for water vapor carried by the monsoon to enter the Tibet Plateau during the ISM period, which could characterize the spread of pollutants from the Indian subcontinent. The concentrations of GEM and PBM**

[revised manuscript text omitted]

-

Zhang, W., Tong, Y., Hu, D., Ou, L., and Wang, X. J. A. e.: Characterization of atmospheric
mercury concentrations along an urban–rural gradient using a newly developed passive
sampler, 47, 26-32, 2012.

**Responses to the Reviewers' Comments**

**First Observation of Mercury Species on an Important Water Vapor Channel in the**

**Southeast Tibetan Plateau**

Dear editor and reviewer,

We greatly appreciate the useful comments and suggestions from the editor and reviewers. We think the novelty and importance of this study have been acknowledged by the reviewers. We have revised the manuscript thoroughly based on the reviewers' comments. Detailed point by point responses are provided below. All the revisions have been highlighted in blue in the revised manuscript. We hope the revised manuscript could meet the standard of ACP. Thanks again for your considerations.

**Anonymous Referee #1**

**Comment #1**

**General comment**

The manuscript by Lin et al. carried out a half-year of continuous measurements of speciated atmospheric Hg as well as a year of measurements of gaseous elemental Hg using a passive sampling technique at a high-altitude station in the eastern Tibetan Plateau. This study combined field observations with backward trajectory analysis, criteria pollutants and a PCA source identification approach, which are used to understand the sources and transport of atmospheric Hg in the eastern Tibetan Plateau. This study is valuable for the atmospheric Hg research topic especially in the pristine Tibetan Plateau where could be potentially impacted by long-range transport of Hg from surrounding anthropogenic Hg source regions. The authors have provided detailed explanations for the variations in the atmospheric Hg, and I broadly agree with the interpretation and hypothesis. The manuscript is overall well organized and written. I therefore suggest a publication of this manuscript in ACP after addressing the following minor to moderate issues.

**Response #1**

We appreciate the reviewer for dedicating time to review our manuscript and provide constructive comments. We have updated the cited references, extended discussion content, redrawn some figures and addressed other concerns from the reviewer in the revised manuscript. All the revisions have been highlighted in blue. Detailed responses to the comments are provided as follows.

**Specific comments**

**Comment #2**

Line 67-68: the authors mentioned that numerous studies have been conducted in Europe and North America. As I know, atmospheric Hg studies in China have also obtained many advances in recent years, which should be also mentioned here (instead of using a citation of mercury emission study in China).

**Response #2**

Thanks for the suggestion. We agree with the reviewer that scientists in China have also conducted studies on the behavior of atmospheric Hg and have obtained many advances in recent years. We have added some literature to the introduction section.

**Comment #3**

Line 80-83: I would suggest to cite Hg emission inventories developed in China and worldwide directly. Note that coal combustion is not the exclusive sources of atmospheric Hg in China.

**Response #3**

Thanks for the suggestion. We have added a sentence to exhibit the Hg emission in Asia, as follow:

'**South Asia, and East and Southeast Asia accounted for 10.1% and 38.6% of global emissions**

**of mercury, respectively (UNEP, 2018; Zhang et al., 2015b). '**

**Comment #4**

Line 84-111: I saw the authors introduced many studies on air pollutants in the Tibetan Plateau, and

I agree this is useful for highlighting the need of the present study. However, I would suggest the authors to make a general description of previous atmospheric Hg studies in the Tibetan Plateau, which would help the authors figure out the knowledge gaps in this study area and strengthen the importance of this study in this research topic.

**Response #4**

Thanks for the suggestion. We have added some general description of previous atmospheric Hg studies in the Tibetan Plateau, as follow: '**In the case of atmospheric Hg, monitoring in marginal**

**areas depicted the basic spectrum of atmospheric Hg in the Tibetan Plateau. Monitoring of**

**atmospheric Hg at Shangri-La, Nam Co, Qomolangma, Mt. Gongga, Mt. Waliguan and Mt.**

**Yulong have illustrated atmospheric Hg concentrations and transport patterns in the Tibetan**

**Plateau from multiple perspectives, all of which also indicated the effects of transboundary**

**transport on the atmospheric Hg concentrations in the Tibetan Plateau (Zhang et al., 2015a;**

**Yin et al., 2018; Lin et al., 2019; Fu et al., 2008; Fu et al., 2012; Wang et al., 2014). For example,**

**our previous study in the QNNP, on the southern border of the Tibetan Plateau, proved that**

**atmospheric Hg from the Indian subcontinent can be transported across high-altitude**

**mountains, and directly to the Tibetan Plateau under the actions of the Indian monsoon and**

**local glacier winds (Lin et al., 2019). Studies of water vapor mercury and wet deposition of Hg**

**in cities such as Lhasa have demonstrated higher concentrations of Hg species than expected**

**(Huang et al., 2015; Huang et al., 2016b; Huang et al., 2016a). But the monitoring of**

**atmospheric Hg speciation is still rare.**'

**Comment #5**

Line 135: is the rain depth at SET station much higher than the mean in the Tibetan Plateau? Could the author tell something regarding the seasonal patterns in rain depth at SET (noticeable difference between the PISM and ISM)?

**Response #5**

Thanks for the suggestion. We have added the precipitation data in the revised manuscript, as follow:

**'The average annual precipitation is approximately 700-1000 mm at the SET station, much**

**higher than the annual precipitation in Tibet (596.3 mm in 2019). The precipitation at the SET**

**station is 47.7 mm during the period of PISM, and is 528.5 mm during the period of ISM in**

**2019.'**

**Comment #6**

Section 2.3: the study by McLagan et al., 2018 (ACP) should be cited. I suggest the author to briefly introduce how to use the passive technique to calculate the atmospheric GEM concentrations. The current information is not very clear to me.

**Response #6**

Thanks for the suggestion. In view of the length of the article, only literature citations are given in the text and no detailed calculations are given. We have added the following information accordingly: **'Hg concentrations in the atmosphere are then calculated from the mass of sorbed**

**Hg according to the equation obtained from our previous work (Guo et al., 2014).'** and **'Similar**

**passive sampling methods for Hg have been widely used worldwide (McLagan et al., 2018).'**

**Comment #7**

Line 202: why did the authors choose a ending height of backward trajectory of 1000 m agl. A height of 1000 m is almost above the PBL.

**Response #7**

Thanks for the comment. The relative high trajectory arrival height was set mainly due to concerns that the complex topography of the Tibetan Plateau might cause significant disruptions to the trajectory. We reviewed the data and found that the average boundary layer height in Nyingchi is

457 m (data from Global Data Assimilation System (GDAS)). In the revised manuscript, we have recalculated all trajectories and redone all the simulations associated with the trajectories. The arrival height was set at 200 m a.g.l., which is about half of the boundary layer height. Considering that a longer simulation time will bring higher trajectories uncertainty, and 120 hours are sufficient for trajectories transmission over longer distances, every backward trajectory was simulated for 120

hours at 3 hours intervals. Also, we examined the effect of arrival height on the trajectories using different arrival heights (20m, 50m, 200m and 500m, respectively) in June 2019. The results showed that the calculated trajectories of the air masses are almost the same when the arrival height is below

500m. The figure below shows the trajectories to Nyingchi in June 2019 with different air masses arrival height. We also added these results in the support information in the revised manuscript.

[Figure]

Figure Trajectories to Nyingchi in June 2019 with different air masses arrival height

We have changed the describe of the backward trajectory simulations in the revised manuscript to make it clear: '**The trajectory arrival height was set to 200 m a.g.l., which is about half of the**

**boundary layer height. We examined the effects of arrival height on the trajectories using**

**different arrival heights (20m, 50m, 200m and 500m, respectively) in June 2019. The results**

**show that the calculated trajectories of the air masses are almost the same when the arrival**

**height is below 500m (Figure S3). Each backward trajectory was simulated for 120 hours at 3**

**hours intervals for GEM, which can cover China, Nepal, India, Pakistan, and the majority of**

**western Asia.**'

**Comment #8**

Line 206-212: the description of PSCF is not clear to me. A least, the authors should mention the arbitrarily set criterions in GEM concentrations used for different sampling period.

**Response #8**

Thanks for the comment and suggestion. The criterion level was set based on the average GEM concentration during the whole monitoring campaign with Tekran instrument. However, we agree with another reviewer that the PSCF analysis does not provide gainful information in this manuscript. So we have decided to delete the PSCF related discussion in the revised manuscript.

**Comment #9**

Line 235-237: rain depth is a good proxy for the changes of monsoons. However, I would suggest the authors to show the air mass sources and transport pathways in different monitoring periods. This would help to better show the changes in monsoons.

Alternatively, the authors may provide the Indian monsoon index to support the changes in ISM.

**Response 9**

Thanks for the suggestion. We have added the Indian Monsoon Index for 2019 in the supporting information (Figure S1), with the Indian monsoon starting to break out in May, 2019 and becoming the dominant wind field. We have also calculated trajectories for different seasons and added a discussion of the sources of trajectories in section 3.3 to calculate transport pathways' changes.

**Comment #10**

Line 395-252: the authors did not show the GEM, GOM and PBM during the ISM3 period. The ISMS is characterized by elevated GEM and decreasing GOM and PBM. Would these observations be explained by wet deposition removal processes?

**Response #10**

Thanks for the comment. We have added a discussion for the ISM3 period. The wet deposition removal process is one of the reasons for the decrease of GOM and PBM, but not the only reason, as GOM and PBM concentrations continue to decline when precipitation declines from ISM2 to ISM3. This may indicate that less GOM and PBM were transported to the SET station or with fewer local sources during ISM3.

**Comment #11**

Line 255: the mean GEM measured by Tekran instrument should be presented.

**Response #11**

Thanks for the suggestion. We have added the mean GEM measured by Tekran instrument here accordingly.

**Comment #12**

Figure 4: this figure contains to many information and I can only read the diurnal GEM trend clearly.

I would suggest to redraw these figures by separating some of the observations in different figures (some maybe in SI). Also, these figures are lacking of Y axis.

**Response #12**

Thanks for the suggestion. We agree with the reviewer that the figures contain too much information.

We have redrawn the diurnal variation figures, keeping only GEM and error range, GOM, PBM and wind speed information.

**Comment #13**

Section 3.2: the authors presented the diurnal patterns in criteria pollutants in Figure 4, but they did not use these data to explain the sources and factors regulating the atmospheric Hg. I would suggest to use the CO (or NO2) to strengthen their hypothesis.

**Response #13**

Thanks for the suggestion. We agree with the reviewer that the use of CO (or $NO_2$) could facilitate the understanding of the changing patterns of GEM. However, the relations between the pollutants and atmospheric Hg are extremely complicated, and due to the word limit, we didn't make very detailed expansion on the manuscript. For example, the relationships between GEM and other pollutants may be significantly affected by the complex topography and precipitation conditions at

Nyingchi. The presence of abundant vegetation may also affect GEM concentrations.

**Comment #14**

Figure 5: these figures are difficult to read. I would suggest the authors to add tables in these figures, which may include the relative fractions, travelling height, mean GEM, GOM and PBM

concentrations for the grouped clusters. Alternatively, they can show these information by text directly in the figures (information using thickness and color of the lines are difficult to obtain)

 **Response #14**

 Thanks for the suggestion. We agree with the reviewer that the figures presented here are difficult

 to obtain useful information. We have redrawn the trajectory and showed detailed information

 concerning the cluster number, GEM concentration and ratio on the trajectory edges by text directly

 in the figures in the revised manuscript. We hope the new version can provide these information

 clearly.

 **Comment #15**

 Line 410-417: would the transport of Hg from southwestern China contribute to the elevated GEM

 during ISM3?

 **Response #15**

 Thanks for the suggestion. The old version trajectories showed that the transport of Hg from

 southwestern China might contribute to the elevated GEM during ISM3. However, after we re-

 calculate the backward trajectories in lower arrival height, we didn't found trajectories from

 southwestern China.

 **Comment #16**

 Section 3.3: the authors mainly use backward trajectories to show the sources and transport

 pathways. I suggest the authors to add an analysis of wind dependence distribution of GEM, GOM,

 and PBM. This would help to support the findings using trajectories (trajectory has many

 uncertainties especially for mountainous monitoring sites.)

 **Response #16**

 Thanks for the suggestion. We agree with the reviewer that trajectory analysis for mountainous

 monitoring sites could be affected and have higher uncertainties. We didn't show the wind

 dependence distributions of GEM, GOM, and PBM in this paper, mainly because of the complex

 topography of the SET station. The final arrival wind direction may be influenced by local

 vegetation or small local topography, and may not reflect the true atmospheric transport trend.

 **Comment #17**

 Line 420: are these PSCF figures showing the sources of GEM, or GOM and PBM? Overall, the authors did not well explain the sources and transformation of GOM and PBM, neither combined them with GEM to propose the atmospheric processes (or sources) of atmospheric Hg in the high- altitude regions.

**Response #17**

Thanks for the suggestion. The PSCF figures show the sources of GEM. As we mentioned above, we agree with another reviewer that the PSCF analysis does not provide gainful information in this manuscript. So we have decided to delete the PSCF related discussion in the revised manuscript.

**Comment #18**

Line 465: would GOM be emitted from land surfaces? The elevated GOM accompanied by increasing solar radiation many indicate in situ oxidation of GEM?

**Response #18**

Thanks for the suggestion. We agree with the reviewer that strong solar radiation in Tibet may indicate in situ oxidation of GEM. We did find that intense solar radiation may be associated with extremely high GOM concentrations. We have added some discussions at the end of section 3.1:

**'Table S3 shows the variations of Hg species, meteorological factors and other pollutants from**

**June 1 to 4, 2019. High GOM concentrations were observed on June 2 and 3, and very high**

**solar radiation and UV Index were also observed in these days. PBM concentrations, relative**

**humidity and $O_3$ were low during this period. The solar radiation was nearly twice the mean**

**value of the ISM1 phase (162.79 W $m^{-2}$, Table S2), and thus higher solar radiation might**

**contribute to the higher GOM concentrations. Some of the PBM might be converted to GOM,**

**but the decrease in PBM concentration was less than the increase in GOM concentration.**

**Generally, high $O_3$ concentrations should be observed at high solar radiation (Kondratyev et**

**al., 1996), but low $O_3$ concentrations were found at Nyingchi, suggesting that $O_3$ may be**

**involved in the formation of GOM. The oxidation of GEM by OH and $O_3$ to generate GOM**

**has been discussed in previous studies with model simulation (Sillman et al., 2007), which may**

**explain the reduced concentration of $O_3$, while OH radicals may be associated with high solar**

**radiation. The mechanism of GOM formation should be further explored in future studies.'**